# The *Eleanor* ncRNAs activate the topological domain of the *ESR1* locus to balance against apoptosis

Mohamed Osama Ali Abdalla [1,2,8], Tatsuro Yamamoto[1,3,4,8], Kazumitsu Maehara[5], Jumpei Nogami[5], Yasuyuki Ohkawa[5], Hisashi Miura [6], Rawin Poonperm[6], Ichiro Hiratani[6], Hideki Nakayama[4], Mitsuyoshi Nakao[1,7,9] & Noriko Saitoh[1,3,9]

MCF7 cells acquire estrogen-independent proliferation after long-term estrogen deprivation (LTED), which recapitulates endocrine therapy resistance. LTED cells can become primed for apoptosis, but the underlying mechanism is largely unknown. We previously reported that *Eleanor* non-coding RNAs (ncRNAs) upregulate the *ESR1* gene in LTED cells. Here, we show that *Eleanors* delineate the topologically associating domain (TAD) of the *ESR1* locus in the active nuclear compartment of LTED cells. The TAD interacts with another transcriptionally active TAD, which is 42.9 Mb away from *ESR1* and contains a gene encoding the apoptotic transcription factor FOXO3. Inhibition of a promoter-associated *Eleanor* suppresses all genes inside the *Eleanor* TAD and the long-range interaction between the two TADs, but keeps *FOXO3* active to facilitate apoptosis in LTED cells. These data indicate a role of ncRNAs in chromatin domain regulation, which may underlie the apoptosis-prone nature of therapy-resistant breast cancer cells and could be good therapeutic targets.

[1]Department of Medical Cell Biology, Institute of Molecular Embryology and Genetics, Kumamoto University, Kumamoto 860-0811, Japan. [2]Department of Clinical Pathology, Faculty of Medicine, Suez Canal University, Ismailia 41522, Egypt. [3]Division of Cancer Biology, The Cancer Institute of JFCR, Tokyo 135-8550, Japan. [4]Department of Oral and Maxillofacial Surgery, Faculty of Life Sciences, Kumamoto University, Kumamoto 860-8556, Japan. [5]Division of Transcriptomics, Medical Institute of Bioregulation, Kyushu University, Fukuoka 812-8582, Japan. [6]Laboratory for Developmental Epigenetics, RIKEN Center for Biosystems Dynamics Research (BDR), Kobe 650-0047, Japan. [7]Core Research for Evolutional Science and Technology (CREST), Japan Science and Technology Agency, Tokyo 104-0004, Japan. [8]These authors contributed equally: Mohamed Osama Ali Abdalla, Tatsuro Yamamoto. [9]These authors jointly supervised this work: Mitsuyoshi Nakao, Noriko Saitoh. Correspondence and requests for materials should be addressed to M.N. (email: mnakao@gpo.kumamoto-u.ac.jp) or to N.S. (email: noriko.saito@jfcr.or.jp)

Breast cancers expressing estrogen receptor-α (ER) depend on estrogen for their growth. ER-positive cancer patients are treated with endocrine therapies that function by blocking estrogen production; these therapies include the use of aromatase inhibitors (AIs)[1]. After treatment, the disease frequently recurs because cancer cells gradually develop tolerance to estrogen deprivation[1–3].

Long-term estrogen deprivation (LTED) cells are established as survivors, after culturing MCF7 ER-positive human breast cancer cells in an estrogen-depleted medium for a long duration[4–7]. LTED cells therefore represent the development of AI resistance or postmenopausal tumorigenesis. Several studies, including the present study, have shown that LTED cells are primed for cell death, which may be triggered by several polyphenols[8,9]. High estrogen concentrations may have a similar effect; it has been shown to be effective against AI-resistant breast cancers, although the treatment is seemingly paradoxical[10,11]. The mechanism underlying LTED cell death may be related to epigenetic chromatin structure, but details of this are largely unknown.

In the nucleus, chromosomes form higher-order structures that govern gene regulation[12,13]. To analyze inter-chromosomal interactions, C-techniques have been developed and shed light on the three-dimensional (3D) genome structures within the nucleus. The original Hi-C analysis classified the chromosome organization into two patterns: transcriptionally active A and inactive B compartments[12]. Subsequent higher-resolution analyses showed that neighboring chromosomal regions were folded into self-associating topologically associating domains (TADs) that are several hundred kb to 1–2 Mb in size[14,15]. In addition, intense point-to-point loops have been detected by high-resolution Hi-C, and some are anchored with the chromatin protein, CTCF[16].

An important recent observation is that the majority of the genome is continuously transcribed into non-coding RNAs (ncRNAs), many of which are specifically transcribed according to cell type or disease[17]. Among them are various types of long ncRNAs including those resulting from divergent transcription at the transcription start sites (TSSs) of protein-coding genes[18–20]. These promoter-associated ncRNAs (paRNAs) exist throughout the genome and positively correlate with proximal gene activities.

Several possibilities of ncRNA-mediated chromatin domain regulation have been suggested[13,21]. ncRNAs that are associated with proteins critical for chromatin domain boundaries, including CTCF and cohesin components, have been identified[22]. ncRNAs that create large RNA foci or clouds in the nucleus regulate gene expression by chromatin domain units[23,24]. Bioinformatics analyses have identified over 700 ncRNA loci that overlap with TAD borders[25]. These results suggest that ncRNAs could be potential therapeutic target candidates[26]. However, the involvement of ncRNAs in chromatin domain regulation in cancer cell adaptation has not been demonstrated yet.

Previously, we and others showed that the gene encoding ER (ESR1), to which cancer cells are transcriptionally addicted, is critically upregulated during LTED adaptation[4,27]. We showed that, in LTED cells, a cluster of ncRNAs, ESR1 locus enhancing and activating non-coding RNAs (Eleanors) were transcribed from a large chromatin domain containing the ESR1 locus and then remained at the transcriptionally active ESR1 locus to form RNA clouds and cis-activate transcription[27]. Eleanors were found to be effectively suppressed with resveratrol due to its estrogenic effect. Eleanors were present in ER-positive breast cancer tissues. However, the mechanisms and significance of the Eleanor-mediated gene and chromatin regulation remained to be investigated.

In the present study, we performed a 4C-Seq (chromosome conformation capture combined with high-throughput sequencing) analysis using the ESR1 promoter as bait and found that Eleanors delineate the TAD of the ESR1 locus. We also found that the identified promoter-associated Eleanor mediates long-range chromatin interaction between the ESR1 and FOXO3 loci, which function in cell proliferation and apoptosis, respectively. Inhibition of Eleanor disrupted the long-range chromatin interaction and suppressed ESR1, but not FOXO3, leading cells to apoptosis. Our observations provide an epigenetic model for the seemingly paradoxical upregulation of the two genes and may explain the apoptosis priming of breast cancer cells undergoing therapeutic estrogen deprivation.

## Results

**Identification of the _Eleanor_ TAD on human chromosome 6q25.1.** To explore the dynamics of higher-order chromosomal organization in breast cancer cells, we used three cellular models: MCF7, LTED, and LTED-RES cells (Fig. 1a). MCF7 cells represent human ER-positive breast cancer. LTED cells were established by culturing MCF7 cells in an estrogen-depleted medium over a long duration (>3 months). At an early stage of estrogen deprivation, cell death occurs because MCF7 cells require estrogen for growth. Those that survive are known as LTED cells and represent breast cancer that has acquired resistance to endocrine therapy[4,28]. To obtain LTED-RES cells, LTED cells were treated with 100 μM resveratrol for 24 h. LTED-RES cells also undergo cell death that could recapitulate estrogen additive therapy because resveratrol and estrogen are structurally related. Previously, we showed that Eleanor nuclear ncRNAs emerged from an approximately 700 kb chromatin region including the ESR1 locus to upregulate ESR1[27]. The resveratrol treatment repressed Eleanors and downregulated ESR1 expression[27].

To investigate the 3D genomic structures of Eleanor-expressing chromosomal regions, we performed 4C-Seq analysis of MCF7 and LTED cells using the TSS of ESR1 as bait. We designed two 4C-Seq sets, one using DpnII (exp-A) and another using HindIII (exp-B) for the first restriction enzyme digestions of the fixed nuclear chromosomes (Supplementary Fig. 1a). The resultant circular DNAs after ligation, which contained genomic sequences fused with the bait, were sequenced and their reproducibility was confirmed between the replicated experiments as well as experiments using different restriction enzymes (Supplementary Fig. 1b, c).

As expected from the nature of C-technology, massive peaks were detected around the bait site (Fig. 1b). Sharp transitions occurred at chr6:151,650,000–151,750,000 and 152,650,000–152,750,000, and their 4C-Seq reads were statistically distinct from those of the neighboring regions (P < 0.05), according to Turkey–Kramer test along chr6: 151,000,000–153,000,000 with 100-kb windows (Supplementary Fig. 2a). We found that the sharp transitions coincided with the TAD boundaries determined by a previous Hi-C study using MCF7 cells (yellow vertical lines in Fig. 1b)[29], as well as those from other Hi-C data of different cell types (yellow vertical lines in Supplementary Fig. 2b, c). The alignment of this TAD and our previous RNA-Seq data[27] revealed that the TAD coincided with the Eleanor-expressing chromatin domain (chr6: 151,750,000–152,750,000) (Fig. 1b). Therefore, the Eleanor TAD resides at 6q25.1 on human chromosome 6 exists in both MCF7 and LTED cells and contains ESR1 and 3 other genes: CCDC170, C6orf211, and RMND1 (Fig. 1b).

To investigate the significance of the Eleanor TAD, we measured the transcriptional activities of genes both within and outside the Eleanor TAD. Quantitative reverse transcription polymerase chain reaction (qRT-PCR) showed that only the genes within the TAD (RMND1, C6orf211, CCDC170, and ESR1) were co-upregulated in LTED cells and cooperatively decreased

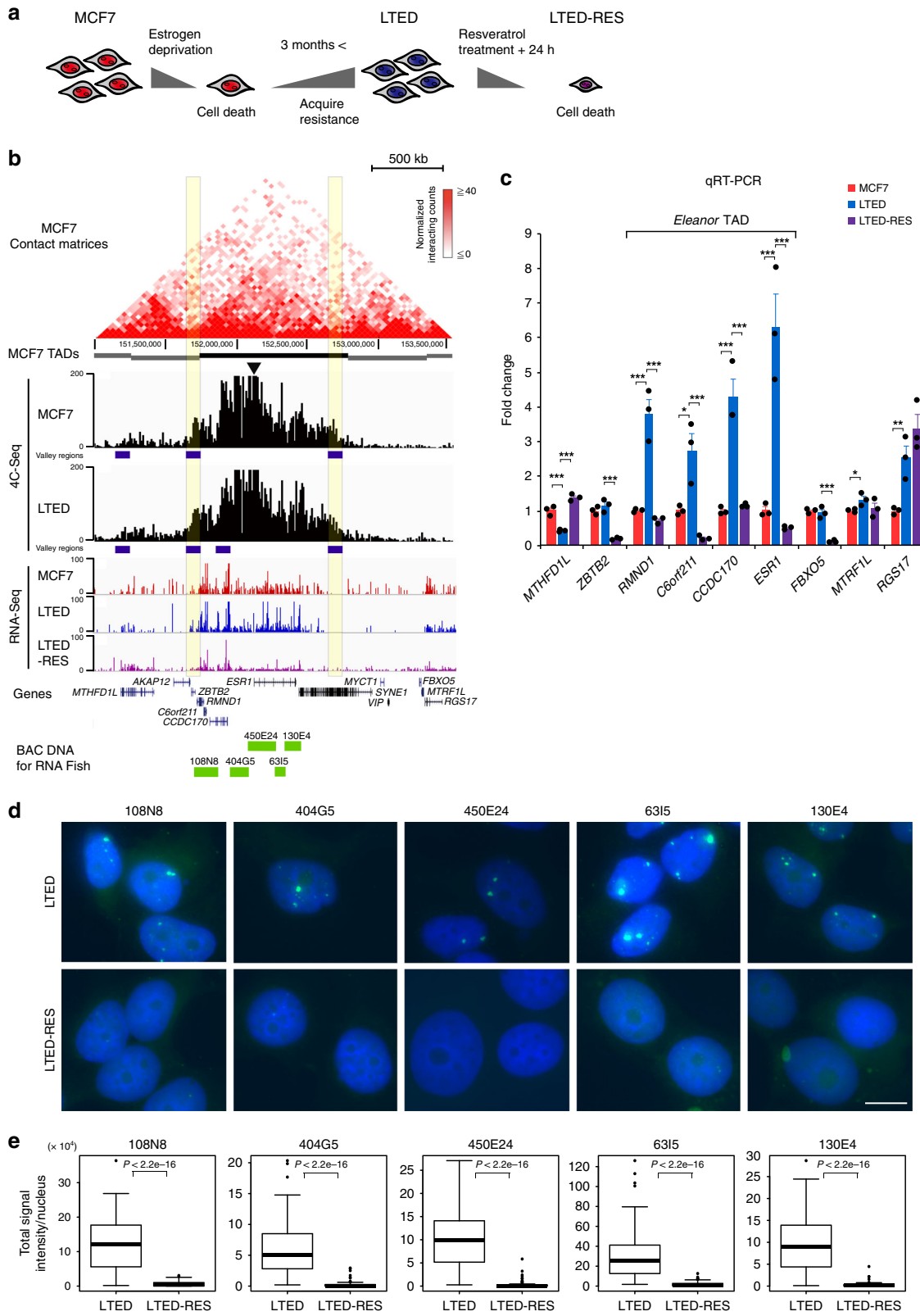

after resveratrol treatment (Fig. 1c). The same TAD-based gene regulation was observed in another ER-positive breast cancer cell line HCC1428 as well as in LTED cells (HCC1428-LTED) and resveratrol-treated cells (HCC1428-LTED-RES) created from the cell line (Supplementary Fig. 2d). Our RNA-Seq of MCF7, LTED, and LTED-RES cell data also support the theory that the four clustered genes are co-regulated in the *Eleanor* TAD

(Supplementary Table 1)[27]. Consistently, the active form of RNA polymerase II was bound to all four genes in the LTED cells, and the binding was reduced in LTED-RES cells (Supplementary Fig. 2e).

*Eleanors*, clusters of ncRNAs that are transcribed from intergenic regions of the *Eleanor* TAD form larger RNA clouds in LTED nuclei, compared with those in MCF7 nuclei[27]. Our

**Fig. 1** *Eleanor* topologically associating domain (TAD) corresponds to the *Eleanor*-expressing chromatin domain. **a** Cells used in this study: MCF7, long-term estrogen deprivation (LTED), and LTED-RES cells. MCF7 cells were cultured in estrogen-deprived medium for >3 months to establish LTED cells, which was then treated with 100 μM resveratrol for 24 h (LTED-RES cells). Cell death occurs early during estrogen deprivation and after resveratrol treatment. **b** Alignment of TADs and 4C-Seq (chromosome conformation capture combined with high-throughput sequencing) profiles in the region, including the *ESR1* gene on human chromosome 6 (6q25.1). Top: Hi-C contact matrix and predicted TAD positions (gray and black bars)[29]. Middle: 4C-Seq (this study) and RNA-Seq[27] profiles of the indicated cells. The arrowhead indicates the position of the 4 C bait, and the dark blue bars indicate the valley regions of the 4C peaks (Supplementary Fig. 2a). Bottom: positions of RefSeq genes and BAC clones (green bars) used as probes for RNA fluorescence in situ hybridization (FISH) in this study. The black bar TAD with yellow highlights delineates the position of the *Eleanor* TAD. **c** Quantitative reverse transcription polymerase chain reaction analysis for the expression levels of genes inside and outside the *Eleanor* TAD. Genes inside the *Eleanor* TAD were cooperatively activated in LTED cells and were downregulated by resveratrol treatment (LTED-RES). The value of MCF7 expression level was set to 1. Data are representative of three independent experiments (mean ± s.e.m.). *P* values were calculated using unpaired, two-tailed, Student's *t* test (\**P* < 0.05, \*\**P* < 0.01, \*\*\**P* < 0.001). **d** RNA FISH analysis along the *Eleanor* TAD. BAC clones used as probes are indicated above each panel. *Eleanor* RNA foci were diminished with resveratrol treatment (LTED-RES). Scale bar, 10 μm. **e** Quantification of RNA FISH. *n* = 50–100 nuclei per sample. Boxplots shown are center line, median; box limits, upper and lower quartiles; whiskers, 1.5× interquartile range; points, outliers. *P* values were calculated using two-tailed, Mann–Whitney *U* test

RNA fluorescence in situ hybridization (FISH) analysis used a series of probes covering the entire *Eleanor* TAD region to show that the cloud was diminished by resveratrol treatment (Fig. 1d, e). These results indicated that resveratrol treatment affects the transcription of all genes in the TAD possibly mediated by *Eleanors*.

**Interaction of the *ESR1* promoter with the *FOXO3* gene locus**. Upon further analysis of our 4C-Seq data, we observed intrachromosomal *ESR1* promoter interactions along the entirety of chromosome 6 in MCF7 and LTED cells (Fig. 2a and Supplementary Fig. 3a). Alignment of the 4C-Seq with MCF7 cells A/B compartment profiles[29] revealed that the *Eleanor* TAD was in the A compartment (Fig. 2a and Supplementary Fig. 3a). 4C-Seq peaks representing interactions with the *ESR1* promoter were prominent near the bait region and gradually reduced in intensity with increased distance from the bait, as expected. However, the 4C peaks tended to coincide with the A compartments throughout the chromosome. In total, 66% and 72% of the regions with 4C peaks overlapped with the A compartments in MCF7 and LTED cells, respectively (Supplementary Fig. 3b). It suggests that the *ESR1* promoter region in the *Eleanor* TAD interacted with genes in the A compartments of multiple regions of chromosome 6. Consistently, *ESR1* promoter-interacting regions showed moderate coincidence with genes that were upregulated in LTED cells (red bars below 4C-Seq profiles in Fig. 2a and Supplementary Fig. 3a). In total, 60% of the regions with 4C peaks in the entire chromosome 6 overlapped with the genes upregulated in LTED cells (Supplementary Fig. 3c).

We then focused on one of the *ESR1* promoter-interacting regions, 6q21, which is approximately 42.9 Mb away from the *ESR1* promoter. First, we analyzed the interaction between the *ESR1* promoter and regions inside (b site) and outside (d site) of 6q21 using DNA FISH (Fig. 2b, c). The frequency of overlapping signals was high at the b site, but not at the d site, in LTED cells (Fig. 2b, c and Supplementary Fig. 3d). Furthermore, 3C–quantitative PCR (3C-qPCR) confirmed that the level of interaction between the *ESR1* promoter and the b site was high in LTED cells, but not at the a, c, or d sites (Fig. 2d). Using genomic qPCR, we confirmed that the increased interaction frequency of the *ESR1* promoter-b site was not due to genome amplification because there was no detectable genome amplification at sites a–d, as with the *ESR1* promoter (Supplementary Fig. 3e).

To understand the significance of genome interaction between the *ESR1* promoter and 6q21, we investigated transcriptional activities at the b site of 6q21. Our qRT-PCR and RNA-Seq showed that most genes at the b site were also upregulated in LTED cells derived from MCF7 (Fig. 2e, Supplementary Table 2)

and HCC1428 cells (Supplementary Fig. 3f). This suggested that, although the two domains are located over 40 Mb apart on chromosome 6, they physically associate in the nuclear space and are co-upregulated in LTED cells.

One of the genes at the b site was *FOXO3*, which belongs to the O subclass of the forkhead family of transcription factors, implicated in the induction of apoptosis via upregulation of genes necessary for cell death such as *Bim* and *PUMA* or downregulation of anti-apoptotic proteins, such as *FLIP*[30,31]. Although the interaction between genes for growth (*ESR1*) and apoptosis (*FOXO3*) appeared contradictory at first, apoptotic activity is frequently enhanced in cancer, both in cultured cell lines and patient cells[6,8,32,33]. During LTED cell establishment, a massive number of cells die immediately after estrogen deprivation commences. Therefore, *FOXO3* may play an important role in cellular adaptation to this environmental change.

To test whether upregulation of *FOXO3* is important for cell death during LTED cell establishment, we knocked down *FOXO3* at an early stage (2 months) of estrogen deprivation and measured cell growth, vitality, and apoptosis (Supplementary Fig. 4a–d). We found that cell death was suppressed by downregulating *FOXO3*, suggesting that *FOXO3* is functionally activated during estrogen deprivation. In addition, we overexpressed *FOXO3* in LTED cells and found that it induced cell death (Supplementary Fig. 4e–g). Knocking down another co-upregulated but irrelevant gene, *SNX3*, did not induce cell apoptosis (Supplementary Fig. 4h, i). Altogether, *FOXO3* is at least partly responsible for cell death during estrogen deprivation, and the surviving LTED cells may overcome its apoptosis-inducing effect, probably by acquiring an ability to upregulate the proliferation gene *ESR1*[27].

**Resveratrol suppressed the interaction between *ESR1* and *FOXO3*.** To investigate whether the chromatin interaction of the *ESR1* promoter was mediated by *Eleanors*, we inhibited *Eleanors* with resveratrol in LTED cells and performed the Hi-C analysis. A comparison of the normalized contact matrices for LTED and LTED-RES showed no dramatic Mb-sized changes (Fig. 3a and Supplementary Fig. 5a). However, local changes were enriched in the A compartments (log$_2$ fold changes in contact (LTED/LTED-RES), shown as blue and red bars below the contact matrix in Fig. 3a), and one was found at the *FOXO3* gene (b site in Fig. 3a). These suggest that the contact between the *ESR1* promoter and *FOXO3* became unstable with resveratrol treatment. Our 3C-qPCR showed that the *ESR1* promoter and the b site were dissociated after resveratrol treatment (Fig. 3b and Supplementary Fig. 5). These results suggest the *Eleanor*-mediated long-range chromatin interaction.

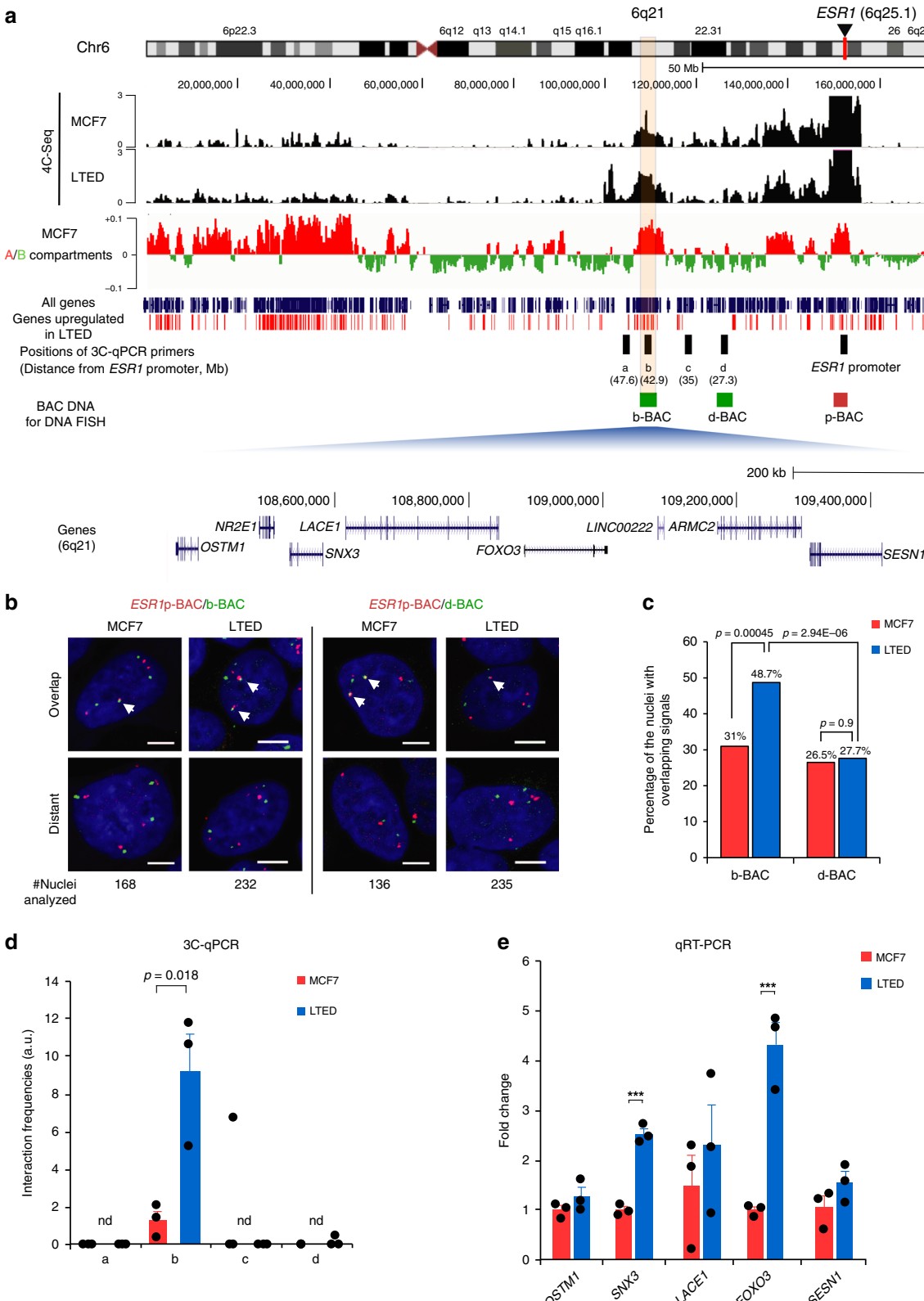

It should be noted that the *FOXO3* gene remained upregulated in LTED-RES (Fig. 3c and Supplementary Fig. 5c), in good agreement with the fact that LTED cells undergo apoptosis upon resveratrol treatment[34]. The expression levels of other genes in 6q21, *SNX*, and *LACE1* were reduced (Fig. 3c, Supplementary Fig. 5c, and Supplementary Table 2). This is not due to RNA stability, because the *FOXO3* RNA was even less stable than

others (Supplementary Fig. 6a). Furthermore, the binding of the active form of RNA polymerase II to the *FOXO3* gene was maintained in LTED-RES cells (Supplementary Fig. 5d).

There are 15 other apoptotic genes that were co-regulated with *FOXO3*. They were upregulated in LTED cells, and their high expression was maintained in LTED-RES cells. (Supplementary Fig. 6b, Supplementary Table 3). Among them, three genes were

**Fig. 2** 4C-Seq (chromosome conformation capture combined with high-throughput sequencing) analysis revealed a long-range interaction between *ESR1* and *FOXO3* genes in long-term estrogen deprivation (LTED) cells. **a** Alignment of 4C-Seq and A/B compartment profiles along the entirety of human chromosome 6 (UCSC genome browser, hg19 assembly). Top: 4C-Seq data showing intra-chromosomal interactions with the *ESR1* promoter used as bait (arrowhead). Middle: Distribution of A (red) and B (green) compartments in MCF7 cells[29]. Note the positive correlation relationship between black peaks from 4C-Seq and red peaks representing A compartments. Bottom: distribution of all genes (blue) and genes upregulated in LTED cells (red)[27]. Thick blue bars represent the positions of 3C-qPCR (chromosome conformation capture–quantitative polymerase chain reaction) primers (a–d and the *ESR1* promoter) used in this study. Green and red bars represent BAC DNA used as DNA fluorescence in situ hybridization (FISH) probes in **b**, **c**. Bottom: RefSeq genes at the b site. It should be noted that the 4C-Seq profiles showed interaction peaks between b site at 6q21 and the *ESR1* promoter (bait for 4C). **b** Representative DNA FISH images of the *ESR1* locus (red, using *ESR1*p-BAC probe) and b and d sites (green, using b-BAC and d-BAC probes, respectively). The position of each probe is shown in **a** (green and red bars). Scale bar, 5 μm. **c** Quantification of the DNA FISH analysis in **b**. Nuclei with overlapping signals at the *ESR1*-b site or *ESR1*-d site were measured. The frequencies of interaction between the *ESR1* and b site increased in LTED cells. *P* values were calculated using two-tailed Fisher's exact test. **d** 3Cq-PCR analysis shows interaction frequencies between the *ESR1* promoter and a–d sites in MCF7 and LTED cells. Positions of sites a–d are shown in **a** (blue bars). **e** Relative expression levels of genes at 6q21 in MCF7 and LTED cells. The MCF7 expression level was set to 1. Data presented in **d**, **e** are representative of three independent experiments (mean ± s.e.m.). *P* values were calculated using unpaired, two-tailed, Student's *t* test (\*\*\**P* < 0.001, n.d. = not detected in at least one experiment)

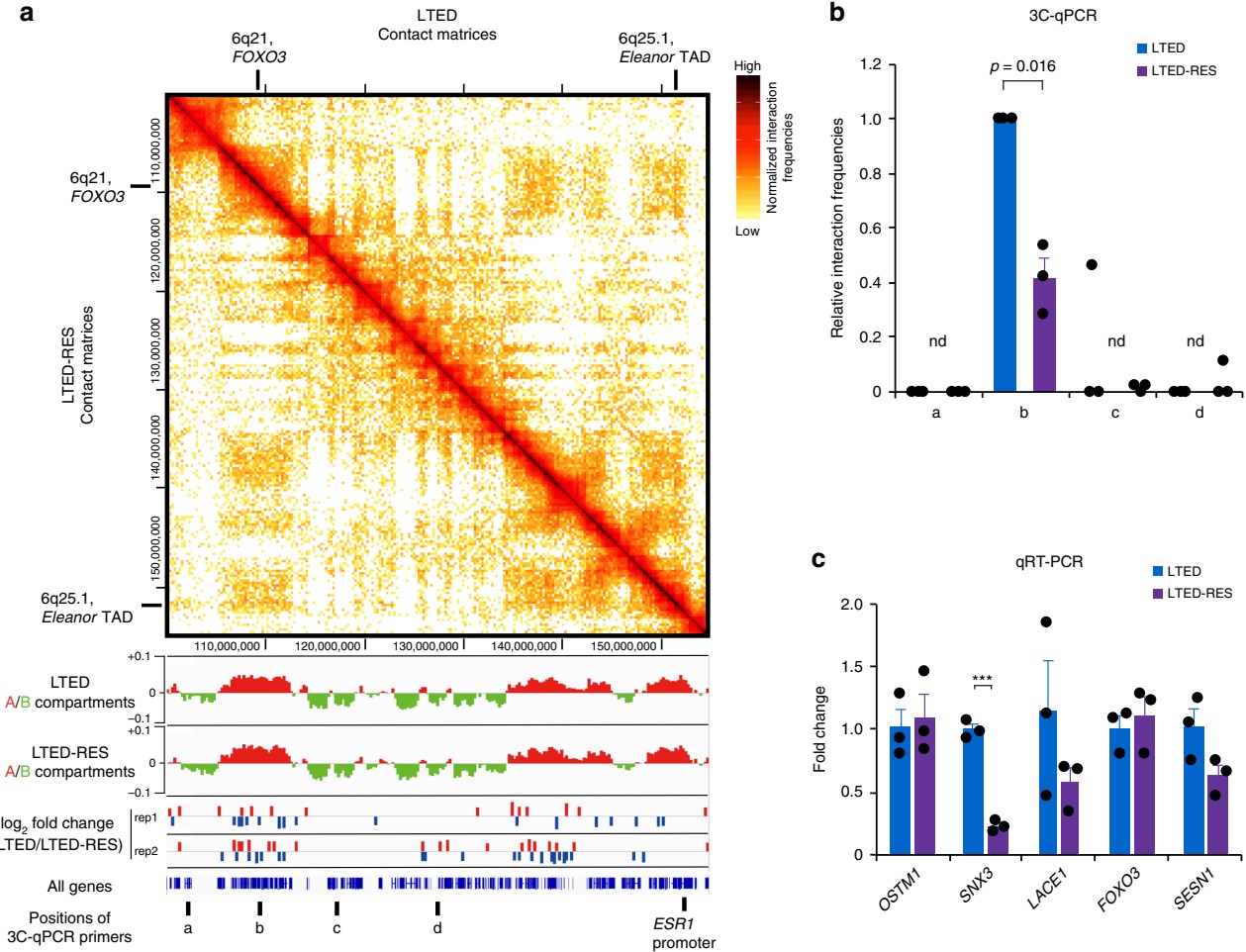

**Fig. 3** Resveratrol treatment reduced the interaction between *ESR1* and *FOXO3* but maintained the *FOXO3* gene activity. **a** The resveratrol treatment affected the interaction between the *Eleanor* topologically associating domain and other A compartments. Top: Contact matrices from Hi-C, binned at 250-kb resolution. The normalized interaction frequencies of long-term estrogen deprivation (LTED) and LTED-RES cells are shown above and below the diagonal line, respectively. Middle: A/B compartments and fold changes in the contact frequencies. Positions of gained (red) and reduced (blue) contacts with the *ESR1* promoter are shown. Bottom: Positions of all genes and 3C-qPCR (chromosome conformation capture–quantitative polymerase chain reaction) primers, as in Fig. 2a. **b** 3C-qPCR shows the relative interaction frequencies of the b site decreased by resveratrol treatment. **c** Relative expression levels of genes at 6q21 in LTED and LTED-RES cells. The LTED expression level was set to 1. Data presented in **b**, **c** are representative of three independent experiments (mean ± s.e.m.). *P* values were calculated using unpaired, two-tailed, Student's *t* test (\*\*\**P* < 0.001, n.d. = not detected in at least one experiment)

encoded on chromosome 6, and all of them were located in the A compartment in LTED cells (Supplementary Table 3). The transcription of *Eleanors and Eleanor* TAD genes may be upregulated through interactions with the active *FOXO3* gene, which is paradoxically required to resist cell death in the transition from MCF7 to LTED.

**The *ESR1* promoter is transcribed in the forward direction**. As we used the *ESR1* promoter as the bait for the 4C-Seq analysis, we next investigated transcriptional condition in the *ESR1* promoter that interacts with other active chromosomal regions. A majority of the transcriptional *cis*-elements are transcribed, including promoter/enhancer RNAs[18,35]. To search for TSS- or promoter-associated RNA (paRNA) in the *Eleanor* TAD, we aligned RNA-Seq and mRNA-Seq in which total RNA and poly (A)$^+$ were used as templates, respectively. We found that the region proximal to the TSS of *ESR1* (chr6: 152,128,354–152,128,780) was highly transcribed in the forward orientation in LTED cells (*pa-Eleanor (S)*; S means sense strand), whereas it was moderately transcribed in the reverse direction in MCF7 cells (*pa-Eleanor(AS)*; AS means antisense strand) (Fig. 4a, b). Both were suppressed by resveratrol treatment (Fig. 4a, b). The transcriptional switching from the reverse (in MCF7 cells) to the forward (in LTED cells) direction was characteristic of the *ESR1* promoter region, because most (90%) of the *ESR1* gene region (chr6:152,128,814–152,424,408) was transcribed in the sense direction. The expression level of *pa-Eleanor* in LTED cells was approximately seven times higher than that in MCF7 and HCC1428 cells (Fig. 4c and Supplementary Fig. 7a). qRT-PCR analysis confirmed that the expression of *pa-Eleanor* was induced in LTED cells and was repressed by resveratrol treatment (LTED-RES) (Fig. 4c and Supplementary Fig. 7a). These observations are consistent with the fact that RNA polymerase II is more highly enriched at the promoter region in LTED cells than in MCF7 cells[27]. Our RT-PCR analysis using a series of primers showed that *pa-Eleanor(S)* is a long ncRNA of at least 400 nt in length and is not contiguous to exon 1 of *ESR1* mRNA (Fig. 4d).

Because *ESR1* is reported to be regulated by ER and FOXO3 transcription factors in breast cancer cell lines[36–38], *pa-Eleanor(S)* levels could be affected by inhibiting their expression. However, qRT-PCR analysis showed that inhibition of ER, either by knockdown of *ESR1* mRNA with RNAi or specific degradation of ER protein with a chemical compound ICI 182,780 (Supplementary Fig. 8b, e), had no effect on *pa-Eleanor(S)* expression (Supplementary Fig. 7b, c). The knockdown of *FOXO3* also had little effect on either the level of *pa-Eleanor(S)* (Supplementary Fig. 7d) or the expression of *Eleanor* TAD genes in LTED cells (Supplementary Fig. 7e).

These results suggested that the amount and direction of transcription at the *ESR1* promoter is highly regulated during the breast cancer adaptation to estrogen deprivation, which may define the level of *ESR1* mRNA.

**pa-Eleanor(S) is required for the *Eleanor* TAD**. To investigate the molecular function of *pa-Eleanor(S)*, we knocked it down in LTED cells using Antisense LNA GapmeR (LNA) (Fig. 5a), which resulted in strong suppression of genes inside the *Eleanor* TAD (*RMND1*, *C6orf211*, *CCDC170*, and *ESR1*) without affecting genes outside the *Eleanor* TAD (*MTHFD1L* and *MTRF1L*) (Fig. 5b). Interestingly, our FISH analysis showed that knockdown of *pa-Eleanor(S)* downregulated the *Eleanor* cloud including itself (450E24) as well as all of the *Eleanors* from the entire 700 kb region (Fig. 5c, d) via an unknown mechanism. These results suggested that *pa-Eleanor(S)* is important for the expression of all *Eleanor* lncRNAs, the formation of the *Eleanor* cloud, and the

activation of genes in the *Eleanor* TAD. These cellular effects were strong, even though the efficacy of the knockdown itself was modest, suggesting the critical regulation of *pa-Eleanor(S)* in the nucleus. Since the results of the *pa-Eleanor(S)* knockdown (Fig. 5a–d) mirrored the effect of resveratrol treatment (LTED-RES), as shown in Fig. 1c, d, *pa-Eleanor(S)* may be a resveratrol target in LTED cells. In fact, in LTED cells, *pa-Eleanor(S)* sharply decreased upon resveratrol treatment (Fig. 4c).

Previously, we identified another *Eleanor* ncRNA, *u-Eleanor*, which activated *ESR1* mRNA transcription and *Eleanor* cloud formation[27]. To understand the regulation mode of *pa-Eleanor* and *u-Eleanor*, we investigated the effects of each knockdown. The *u-Eleanor* knockdown did not change the expression of *pa-Eleanor* (Supplementary Fig. 7f). In contrast, the *pa-Eleanor(S)* knockdown suppressed the *u-Eleanor* expression (Supplementary Fig. 7g), suggesting that *pa-Eleanor* upregulates the *ESR1* transcription through *u-Eleanor* or may directly affect the expression of the *ESR1* mRNA (Supplementary Fig. 7h).

We wondered whether the effects of *pa-Eleanor(S)* knockdown affected events downstream of the reduced *ESR1* mRNA or ER protein level and also whether knockdown of any *Eleanor* TAD RNA product could confer the same effects. To investigate these, we first knocked down *ESR1* mRNA in LTED cells and found that the transcriptional levels of genes in the *Eleanor* TAD, including *pa-Eleanor(S)* and the *Eleanor* cloud, did not change (Supplementary Fig. 8a–d). Therefore, *Eleanors* are specific ncRNAs that play a role in the activation of the chromatin domain. Consistent with this result, ICI 182,780 treatment did not change the expression levels of either *Eleanor* TAD genes or *pa-Eleanor(S)* (Supplementary Figs. 7c and 8e, f). Our previous reports also showed that the *Eleanor* cloud did not disappear after ICI 182,780 treatment[27]. Taken together, these results indicated that activation of the *Eleanor* TAD is not mediated by ER protein; rather, it depends on *Eleanors*.

Since *Eleanors* are important for LTED cell proliferation[27], we wondered whether inhibition of *pa-Eleanor(S)* influenced cell growth. We found that *pa-Eleanor(S)* knockdown suppressed cell proliferation in both a time- and dose-dependent manner (Fig. 5e and Supplementary Fig. 8g). The results were similar to those obtained from resveratrol treatment of LTED cells, again suggesting that *pa-Eleanor(S)* is the target of resveratrol. LNA *pa-Eleanor(S)* specifically targets the forward-oriented transcript, which is predominant in LTED cells. Therefore, cell proliferation was inhibited more strongly in LTED than in MCF7 cells (Fig. 5e and Supplementary Fig. 8g). As the LTED cell formation recapitulates the development of endocrine therapy resistance, LNA for *pa-Eleanor(S)* may be a promising therapeutic agent.

**pa-Eleanor(S) mediates the long-range chromatin interaction**. To investigate the function of *pa-Eleanor(S)* in gene regulation through chromatin structures, we performed 3C-qPCR analysis. We found that the interaction between the *ESR1* promoter and the b site at 6q21 reduced with *pa-Eleanor(S)* knockdown in LTED cells (Fig. 6a), supporting the idea that ncRNA was involved in the long-range chromatin interaction. In contrast, transcriptional levels of the genes at 6q21 showed little-to-no change in the LTED cells with *pa-Eleanor(S)* knockdown (Fig. 6b). Knockdown of *ESR1* mRNA or ER protein degradation with ICI 182,780 also did not cause any significant changes in gene expression in the 6q21 domain (Supplementary Fig. 9a, b). These results indicated that the ER transcription factor contributes very little to the *ESR1* and *FOXO3* genes in LTED cells. More importantly, activation of the *FOXO3* gene in the 6q21 domain is autonomous and not dependent on a long-range chromosomal interaction via *Eleanors*, whereas transcriptional

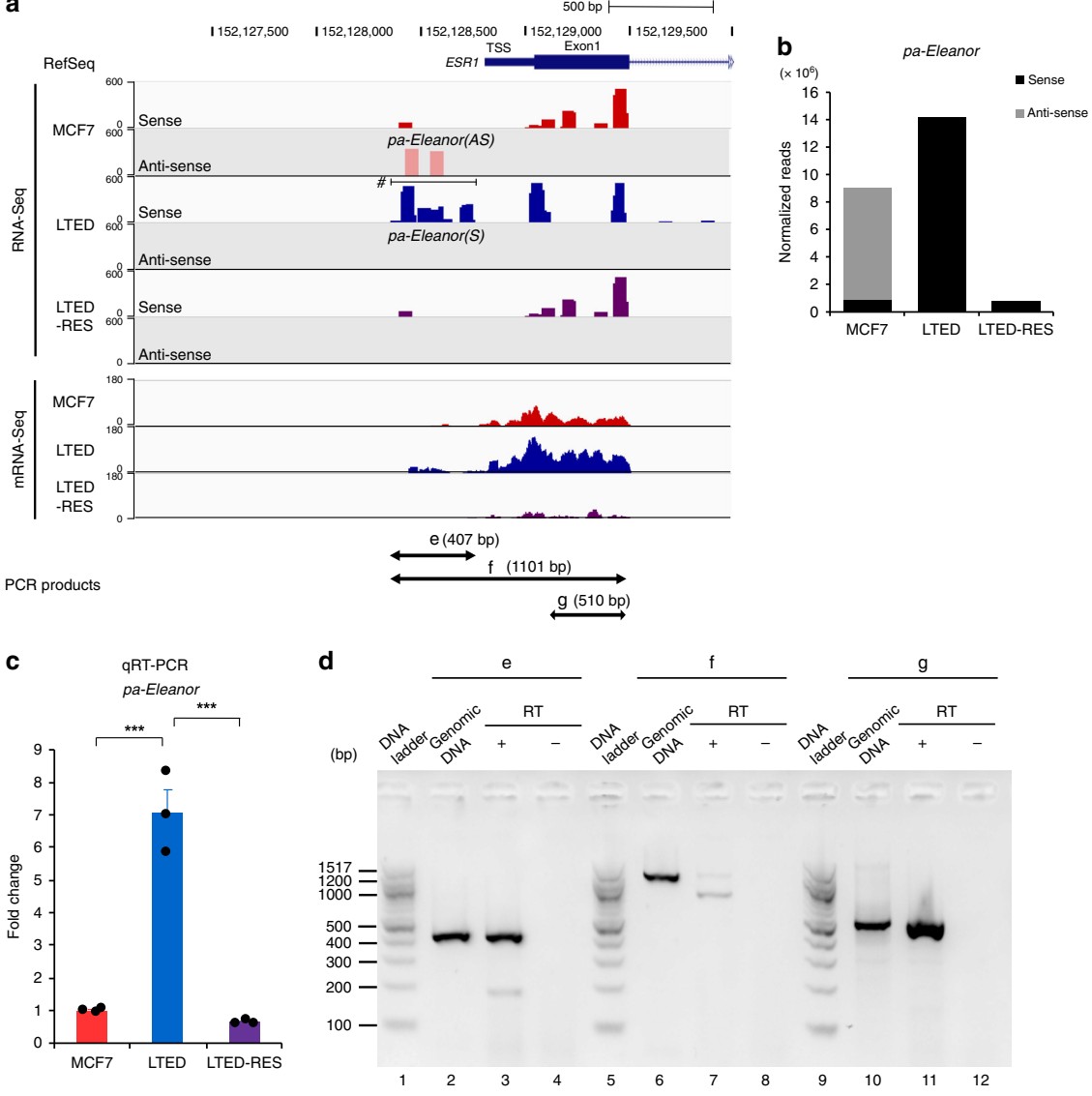

**Fig. 4** The *ESR1* promoter is transcribed differentially in MCF7 and long-term estrogen deprivation (LTED) cells. **a** Alignment of RNA-Seq and mRNA-Seq of the human *ESR1* promoter-proximal region in the indicated cells[27]. Peaks in RNA-Seq indicate transcripts with sense (dark colors) and antisense (pale colors) orientation relative to the *ESR1* gene. In MCF7 cells, antisense transcription was detected at the transcription start site (TSS), which then switched to opposite direction to produce *pa-Eleanor(S)* (denoted by the horizontal bar with hash (#)) in LTED cells. Positions of the PCR products (e–g) in **d** are shown at the bottom. **b** Number of sequencing reads mapped at the TSS of the *ESR1* gene. Numbers corresponding to sense and antisense RNA were counted and normalized according to the total number of mapped reads. **c** Expression levels of *pa-Eleanor* in the indicated cells determined with quantitative reverse transcription polymerase chain reaction (qRT-PCR). Values in MCF7 cells were set to 1. Data are representative of three independent experiments (mean ± s.e.m.). *P* values were calculated using unpaired, two-tailed, Student's *t* test (***$P < 0.001$). **d** Local expression of *pa-Eleanor(S)* at site e in LTED cells. RT-PCR to amplify predicted transcripts as shown in **a** (e–g) resulted in successful detection of the e-fragment (*pa-Eleanor*, lane 3) and g-fragment (exon1 of *ESR1*, lane 11) by electrophoresis but not the f-fragment (lane 7). These results indicate that *pa-Eleanor(S)* is not contiguous with the *ESR1* mRNA. All primers and amplification conditions were validated with PCR in parallel with genomic DNA as a template and RT-PCR with and without reverse transcription (RT+ and −)

activities in the *Eleanor* TAD are dependent on this interaction. Consistent with this high level of *FOXO3* transcription, our Annexin V staining and VB48 assays showed that LTED cells underwent apoptosis upon knockdown of *pa-Eleanor(S)* (Fig. 6c and Supplementary Fig. 10a–c). These results revealed a balanced chromosomal interaction between genes for cell proliferation (*ESR1*) and death (*FOXO3*) in LTED cells, mediated by *Eleanors*. Our data elucidated an ncRNA-mediated mechanism of chromatin interaction that underlies the characteristics of breast cancer cells primed for death.

## Discussion

In this study, we found that *pa-Eleanor(S)* activates the genes in the *Eleanor* TAD and the *Eleanor* cloud formation in LETD cells (Fig. 7a). *ESR1* and *FOXO3*, the genes for proliferation and apoptosis, respectively, are in close proximity and were paradoxically co-upregulated in LTED cells via *Eleanors* (Fig. 7b, left). Upon the inhibition of *pa-Eleanor(S)* by either knockdown or resveratrol treatment, *Eleanors* disappeared and the genes in the *Eleanor* TAD were repressed, whereas the *FOXO3* activity was maintained and induced cell death (Fig. 7b, right). These findings

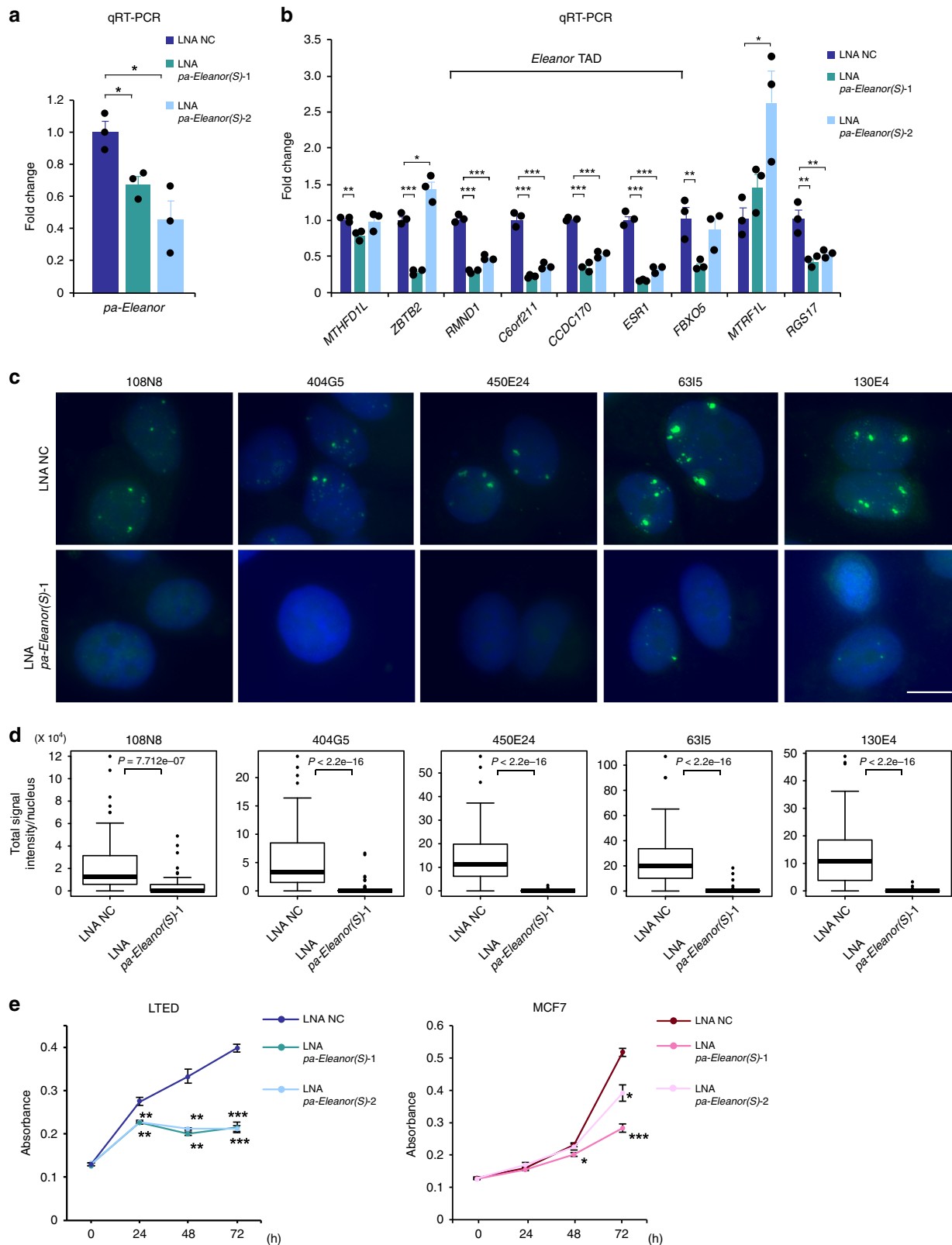

may represent the molecular mechanism underlying the apoptosis-prone nature of endocrine therapy-resistant breast cancer cells.

We propose that the primary function of *Eleanors* is the activation of the *ESR1* gene and neighboring genes in the *Eleanor* TAD (Fig. 7a). The long-range chromatin interaction between *ESR1* and *FOXO3* may also be mediated directly by *Eleanors*. Alternatively, the interaction may be the consequence of the

activation of the *Eleanor* TAD. Upon *Eleanor* inhibition, *ESR1* is inactivated and the interaction with *FOXO3* is destabilized, but both genes are in the A compartment. The genes in the *Eleanor* TAD were repressed while they remained in the A compartment (Fig. 3a). This local chromatin re-wiring makes it possible to maintain the activity of *FOXO3*, while *ESR1* is inactivated. Our findings provided an insight into the regulation of a chromatin

**Fig. 5** *pa-Eleanor(S)* plays a role in *Eleanor* topologically associating domain (TAD) formation and long-term estrogen deprivation (LTED) cell proliferation. **a** Quantitative reverse transcription polymerase chain reaction showed that *pa-Eleanor(S)* was knocked down with Antisense LNA GapmeR (LNA) in LTED cells. **b** Expression levels of genes in *Eleanor* TAD decreased upon *pa-Eleanor(S)* knockdown in LTED cells. Values of LNA for the negative control (LNA NC) were set to 1. **c** RNA fluorescence in situ hybridization (FISH) scanning analysis along the *Eleanor* TAD, with *pa-Eleanor(S)* knockdown in LTED cells. BAC clones used as FISH probes are indicated on the panels. The genomic positions covered by the BAC clones are shown in Fig. 1b (green bars). Nuclear *Eleanor* RNA foci were diminished with *pa-Eleanor(S)* knockdown. Scale bar, 10 μm. **d** Quantitative analysis of RNA FISH in **c**. Total signal intensities per nucleus in LTED cells were measured (*n* > 40 nuclei per sample). Boxplots shown are center line, median; box limits, upper and lower quartiles; whiskers, 1.5× interquartile range; points, outliers. *P* values were calculated using two-tailed, Mann–Whitney *U* test. **e** Inhibition of LTED cell proliferation due to *pa-Eleanor(S)* knockdown. Time course analysis was performed after treatment of LTED (left) and MCF7 (right) cells with LNA. Data presented in **a**, **b**, **e** are representative of three independent experiments (mean ± s.e.m.). *P* values were calculated using unpaired, two-tailed, Student's *t* test (*$P < 0.05$, **$P < 0.01$, ***$P < 0.001$)

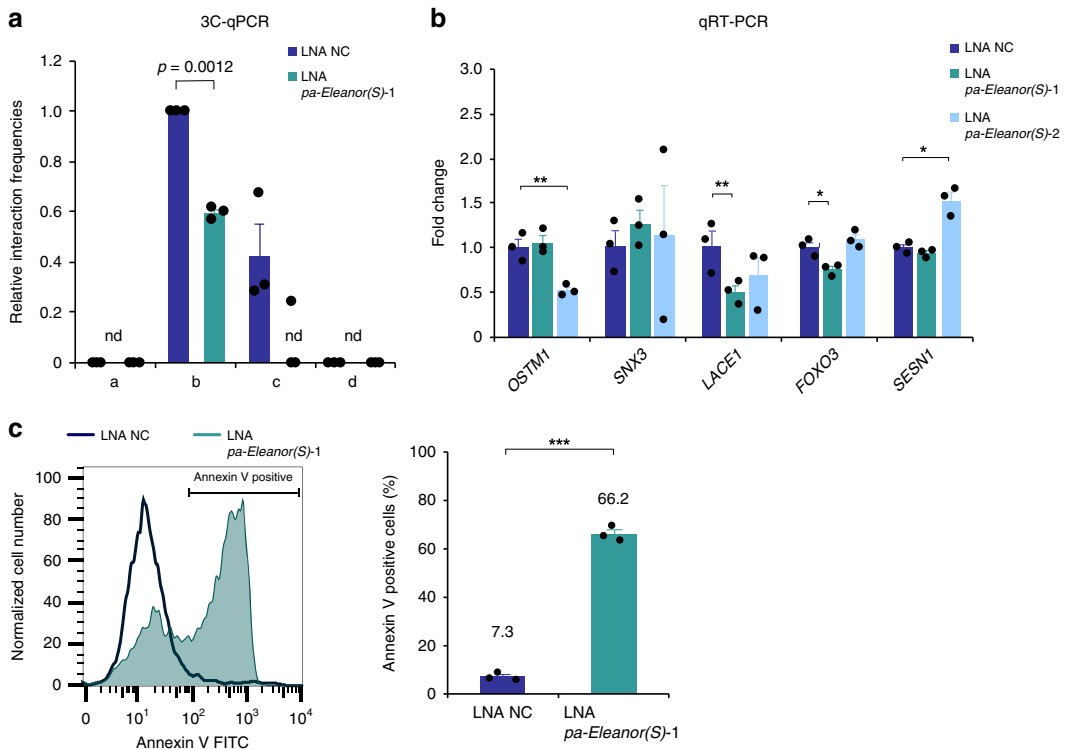

**Fig. 6** Knockdown of *pa-Eleanor(S)* suppressed the long-range chromatin interaction without reducing *FOXO3* gene activity and induced apoptosis in long-term estrogen deprivation (LTED) cells. **a** 3C-qPCR (chromosome conformation capture–quantitative polymerase chain reaction) analysis shows that *pa-Eleanor(S)* knockdown reduced interaction between *ESR1* and b sites at 6q21 in LTED cells. Interaction frequencies were measured between the *ESR1* promoter and indicated positions (a–d; thick blue bars in Fig. 2a). **b** Expression levels of genes around the *Eleanor* topologically associating domain with *pa-Eleanor(S)* knockdown in LTED cells. Quantitative reverse transcription polymerase chain reaction values of cells treated with Antisense LNA GapmeR for the negative control were set to 1. **c** Fluorescence-activated cell sorting (FACS) analysis of Annexin V-stained cells shows that knockdown of *pa-Eleanor(S)* increased apoptosis in LTED cells. A representative FACS profile is on the left. Quantification of the FACS analysis is on the right. Data presented in **a**–**c** are representative of three independent experiments (mean ± s.e.m.). *P* values were calculated using unpaired, two-tailed, Student's *t* test (*$P < 0.05$, **$P < 0.01$, ***$P < 0.001$, n.d. = not detected in at least one experiment)

domain that underlies cancer fragility, in which the genes for cell growth and death interact via paRNA, and suggested that *pa-Eleanor(S)* can be a potential therapeutic target for endocrine therapy-resistant breast cancer cells.

We found that *Eleanor* TAD was in the A compartment and also interacted with other A compartment TADs (Fig. 2). This is consistent with previous reports that describe how compartments of the same type are clustered in the nucleus[12,39]. Mechanisms for TAD formation involving chromatin structural proteins such as CTCF, cohesin, and the cohesin-loading factor Nipbl have been suggested, and local loop extrusion of a chromosome via these factors may be important[16,40–43]. According to publicly available data[44], these factors bind at the boundary of the *Eleanor* TAD,

but they are not particularly enriched relative to regions within the TAD. Rather, we found that transcription of a cluster of ncRNAs (*Eleanors*) delineated the TAD in the A compartment in LTED cells. Other studies have suggested involvement of ncRNAs in TAD regulation[13,42]; however, the question of how general they are remains to be investigated. Because inhibition of *Eleanors* either by resveratrol treatment or specific knockdown decreased transcription of the genes inside the *Eleanor* TAD (Figs. 1c and 5a, b), the RNA products and RNA cloud formation may be critically important for active chromatin domain formation, although other factors including the ncRNA transcription process itself or DNA–RNA hybrid (R-loop) formation may also be important contributing factors.

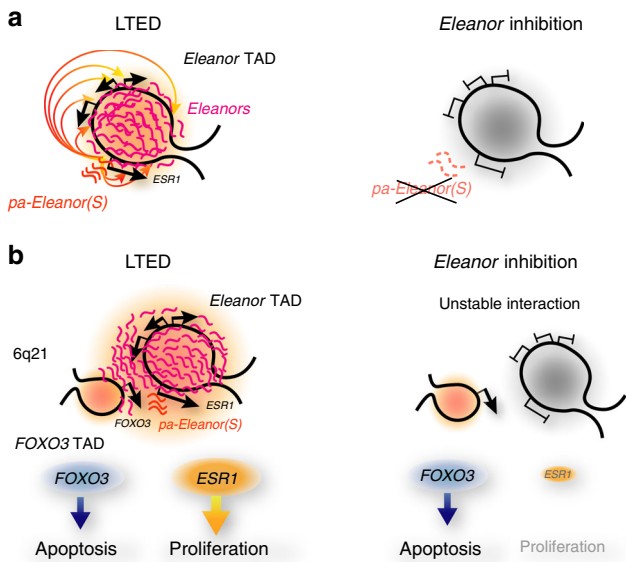

**Fig. 7** Proposed model of *Eleanor* functions in long-term estrogen deprivation (LTED) cells. *Eleanors* play roles in activating genes within the *Eleanor* topologically associating domain (TAD) (**a**) and in mediating the long-range chromatin interaction (**b**). **a** In LTED cells, pa-*Eleanor(S)* activates the transcription of *Eleanors* and RNA cloud formation, thus activating the genes in the *Eleanor* TAD, including *ESR1* (left). *Eleanor* inhibition represses the *Eleanor* TAD (right). **b** In LTED cells, the gene for proliferation (*ESR1*) and the gene for apoptosis (*FOXO3*) are in close proximity, as both are active in the A compartment. Upon inhibition of *Eleanors* by resveratrol or pa-*Eleanor(S)* knockdown, the long-range chromatin interaction decreased and the *ESR1* gene was repressed, but the *FOXO3* gene activity was maintained at a high level. These suggest a mechanism underlying the apoptosis-prone nature of LTED cells

Promoters are the key regulatory elements in long-range chromatin interactions[45]. In this study, we found that the *ESR1* promoter interacted with *FOXO3*, a gene approximately 42.9 Mb away from *ESR1*. We also found that the *ESR1* promoter is bidirectionally transcribed in MCF7 cells, which is similar to other promoters exhibiting divergent transcription[18–20]. As *ESR1* transcription was upregulated, the sense-stranded transcript (pa-*Eleanor(S)*) became more prevalent, which may be characteristic of active promoters[18]. Although the mechanism for the transcriptional switching is unknown, it is possible that RNA polymerase II transcription in the opposite direction may interfere with the movement of RNA polymerase II toward the *ESR1* gene in MCF7 cells, which could be relieved in LTED cells. This idea is shared with transcriptional interference in yeast and other eukaryotes[46,47].

It is clear that a long-range chromatin interaction between *ESR1* and *FOXO3* was enhanced by increased levels of pa-*Eleanor(S)* (Figs. 2d and 4c) and suppressed by inhibition of pa-*Eleanor(S)* (Figs. 3b, 4c, 5a, and 6a). This is the first demonstration of paRNA being responsible for a long-range chromatin interaction and transcriptional activation of genes in the TAD, probably through RNA cloud formation. An intriguing question is whether pa-*Eleanor(S)* shares molecular features with other ncRNAs involved in inter-chromosomal interactions, including *HOTAIR* and *Firre*[48,49].

The present study showed that inhibition of pa-*Eleanor(S)* suppressed *ESR1* and the neighboring genes but did not affect the active gene *FOXO3* (Figs. 5a, b and 6b), which promoted apoptosis in LTED cells. These findings suggested that activation of *ESR1* is dependent on chromosomal interaction with the active *FOXO3* gene via pa-*Eleanor(S)* and the *Eleanor* cloud in LTED

cells. Although the possibility of the mutual activation of the two genes through the chromatin interaction cannot be excluded (Fig. 7b), it is more likely that *FOXO3* is autonomously upregulated and provides the site of active transcription for *ESR1*. Therefore, disconnection of the physical interaction by *Eleanor* inhibition only affected genes in the *Eleanor* TAD, but not *FOXO3*, and could induce apoptosis (Figs. 5a, b and 6b, c). Because targeting of paRNA with antisense oligonucleotides, including LNA, is sequence-specific, it may be a good therapeutic strategy in comparison to targeting other epigenetic factors including histone modifications and their reader proteins that are distributed throughout the genome. Importantly, targeting of paRNA is also strand-specific, giving it an advantage over genome editing, which is not strand-specific.

ER and FOXO3 are intimately and paradoxically linked in breast cancer. They are co-upregulated in breast cancer tissues[33,50] but have opposite biological effects. Estrogen induces transcriptional activity of ER while inactivating FOXO3[51]. ER promotes estrogen-dependent breast cancer cell proliferation and tumorigenesis, whereas FOXO3 suppresses these effects[38]. Our apoptosis analysis showed that FOXO3 is responsible for apoptosis under estrogen deprivation (Supplementary Fig. 4c, d). Another study showed that the antidiabetic drug metformin induces apoptosis by *FOXO3* upregulation in MCF7 cells[52]. *FOXO3* binds to and activates the *ESR1* promoter to induce ER expression in breast cancer[37,53,54]. Furthermore, FOXO3 binds to ER protein in the nucleus and inhibits its transcription factor activity[38,55]. The present study indicated another type of crosstalk in which *ESR1* and *FOXO3* interact and are co-upregulated in the nucleus of LTED cells, which serve as a model for endocrine therapy-resistant breast cancer.

Co-upregulation of genes with seemingly counteracting functions may indicate the robustness and fragility of cancer. Cancer cells survive under various environmental changes via the regulation of gene expression, which allows them to acquire appropriate phenotypes[56]. For LTED cells undergoing unfavorable environmental changes (e.g., estrogen deprivation), it may be inevitable that *FOXO3* is upregulated so that cells become primed for apoptosis[8,9,57], representing the fragility of cancer. In contrast, *ESR1* is upregulated for survival by interacting with the transcriptionally active *FOXO3* gene through *Eleanors*, which represents cancer robustness. In other words, the majority of the LTED cells undergo cell death, at least partly due to upregulation of the apoptotic gene *FOXO3*, as shown in Supplementary Fig. 6. Cells may survive if the proliferation gene *ESR1* acquires the ability to interact with the active TAD, including *FOXO3*, via induction of *Eleanors*. Therefore, an intricate balance exists in LTED cells between apoptosis and proliferation. Targeting chromatin interaction disrupted this sensitive balance and made the cells prone to apoptosis (Fig. 6).

Resveratrol induces apoptosis in LTED cells[34]. The present study showed that pa-*Eleanor(S)* is a target of resveratrol (Fig. 4c). Because resveratrol and estrogen are structurally related, the molecular mechanism presented in this report may explain the efficacy of estrogen additive therapy following AI treatment[8,9,11]. Our study reveals a chromatin interaction mechanism via ncRNAs and sheds light on cancer robustness and fragility.

## Methods
**Cell culture**. MCF7 (ATCC) and HCC1428 (ATCC) cells were cultured in RPMI 1640 (Sigma) supplemented with 10% fetal bovine serum (FBS; Corning). For LTED cells, MCF7 or HCC1428 cells were grown in phenol red-free RPMI 1640 (Wako) containing 4% dextran-coated charcoal-stripped FBS (Thermo Fisher Science) for 3–8 months. LTED cells were treated with 100 μM resveratrol (Sigma) for 24 h (LTED-RES cells) or with 100 nM ICI 182,780 (Tocris) for 48 h (LTED-ICI cells). Cells were cultured at 37 °C and 5% $CO_2$. During the course of this study, all cell lines were tested for mycoplasma contamination.

**4C template preparation**. The experimental strategy for 4C analysis[58] is shown in Supplementary Fig. 1. In brief, at least $10^7$ MCF7 or LTED cells were harvested and crosslinked with 1% formaldehyde for 5 min with rotation at room temperature, followed by quenching with a final concentration of 125 mM glycine and centrifugation for 10 min at $400 \times g$ and 4 °C. Nuclei were digested overnight with the first restriction enzyme at 37 °C, followed by proximity ligation with T4DNA ligase (TaKaRa, 2011A) for 4 h at 16 °C and then kept for 30 min at room temperature. Chromatin was de-crosslinked with 2% proteinase K in an overnight digestion at 65 °C. DNA was extracted with phenol–chloroform and precipitated with ethanol and then further digested with the second restriction enzyme overnight at 37 °C and subsequently ligated in T4 DNA ligase. Resultant DNA circles are 4C templates, which were precipitated in ethanol with glycogen as a carrier (20 μg/mL). The interacting DNA segments captured were amplified with PCR using LA Taq (TaKaRa, RR002A). For 4C experiments, two independent sets (exp-A and exp-B) were performed using different restriction enzymes and primers. For exp-A, DpnII (NEB R0543M) was used as the first restriction enzyme and Csp6I (Thermo, ER0211) as the second restriction enzyme. Primers for 4C library construction for exp-A were as follows: DpnII forward: 5′-TCTGGAAAGACGTTCTTGATC-3′; Csp6I reverse: 5′-CTGGATGAATGGCAGTGG-3′. For exp-B, HindIII (TaKaRa, 1060AH) was used as the first restriction enzyme and DpnII (NEB R0543M) as the second restriction enzyme. Primers for 4C library construction for exp-B were as follows: HindIII forward: 5′-AACATAACCTCAGGTCACGAAC-3′; DpnII reverse: 5′-GCACATAAGGCAGCACATTAG-3′. Each experimental set was repeated for two independent replicates.

**4C-Seq and data analysis**. The 4C-Seq library was prepared using a NEBNext Ultra DNA Library Prep Kit, according to the manufacturer's protocol, and sequenced using the Illumina HiSeq1500 system (Illumina, San Diego, CA, USA). The sequenced reads were mapped to the human reference genome (hg19) using the Bowtie 2 software (v2.1.0) with default parameters[59]. Uniquely mapped reads were selected. PCR duplicates were removed from mapped reads using SAMtools (v0.1.18)[60]. The mapped reads in every 500-kb non-overlapping genomic segment (bin) were counted, and the counts were normalized as read per million (RPM) scores. The differences in RPM scores between 4C and input control data ($RPM_{4C} - RPM_{input}$) were calculated as 4C signal intensity of each bin. Furthermore, we analyzed our 4C-Seq by using the 4C pipeline developed by Tanay and de Laat groups[61].

For Supplementary Fig. 2a, we performed analysis of variance followed by Tukey–Kramer test to calculate the adjusted $P$ values and determined the valley regions in the 4C-Seq peaks. The calculation was done for 20 bins of each 100 kb along the region of chr6: 151,000,000–153,000,000. Prior to the calculation, the 4C-seq reads were normalized at every 10 kb.

**Hi-C experiment**. Hi-C experiments[62] using LTED and LTED-RES cells were performed as follows. Briefly, cells were crosslinked with 1% formaldehyde for 10 min at room temperature, followed by quenching with a final concentration of 125 mM glycine. In total, $1–2 \times 10^6$ fixed cells were used for the following steps. Nuclei were extracted and digested overnight with 200 U DpnII (NEB, R0543S) at 37 °C while shaking (Eppendorf ThermoMixer® C). Next, the nuclei were collected by centrifugation and washed with $1 \times$ NEB buffer 2 and then incubated in Fill-in Mix [including Biotin-14-dATP (Invitrogen, 19524-016), dGTP, dTTP, and dCTP with 15 U of Klenow (NEB, M0210L), 1% Triton X-100)] for 1 h at 25 °C. The nuclei were then collected by centrifugation and incubated in Ligation Mix [$1 \times$ T4 DNA ligase buffer (NEB, B0202S) with 1800 cohesive-end ligation units (CEU) of T4 DNA ligase (Invitrogen, 15224-025), 1% Triton X-100] for 6 h at 16 °C while shaking. Samples were treated with RNase A and were de-crosslinked with proteinase K overnight at 65 °C. DNA was extracted with phenol–chloroform and precipitated with ethanol. Purified DNA samples were checked for quantity and quality by the Qubit dsDNA HS Assay Kits (Thermo Fisher Scientific) and TapeStation 4200 (Agilent), respectively.

For next-generation sequencing libraries, Hi-C DNAs were mixed with T4 DNA pol mix [including dATP, dGTP, and 3 U of T4 DNA polymerase (NEB, M0203S)] and incubated for 30 min at 37 °C. These DNAs were sonicated (150–400 bp) using the DNA Shearing System S220 (Covaris) and size-selected using AMPure XP beads (Beckman Coulter). The size-selected DNAs were captured with Dynabeads® M-280 Streptavidin (Thermo Fisher Scientific). For the sequencing library preparations, a KAPA LTP Library Preparation Kit (KAPA Biosystems) was used, and the purification was performed with AMPure XP beads. The quality of the libraries was monitored with TapeStation 4200, and paired-end sequencing (150-bp read length) was performed using Hiseq X Ten (Illumina). We obtained ~70 million paired-end reads per sample.

**Hi-C analysis**. For the LTED and LTED-RES Hi-C data, the paired-end reads were individually mapped to the human genome (hg19) using the Juicer (v1.5) pipeline[63]. We used mapped reads with MAPQ > 30 to generate.hic files for further analyses. For generating A/B compartment profiles at 250-kb resolution, the first eigenvector of the Pearson's correlation matrix was used[64]. For the analysis of the differences in the Hi-C contact frequency, we extracted the number of interactions (i.e., reads) revealed by Hi-C for every 250-kb bin using the Juicer tools[63] from

the.hic file, and the differences in the interaction frequencies on chr6 (chr6:100,000,000–155,000,000) were calculated using the R package *HiCcompare*[65]. The normalized interaction frequencies and fold-change differences were log transformed and displayed as a heat map, using the R package *ggplot2*. Changes in interaction frequencies with the *Eleanor* TAD were computed, and significant changes were determined using *HiCcompare*.

According to the method developed by the Yue and colleagues[66], we created the Binary Upper Triangular Matrix (BUTLR) File Format, using the published normalized Hi-C matrix data of MCF7 at 40-kb resolution (GEO accession: GSE66733, Supplementary file)[29] and visualized it as a contact matrix on a 3D Genome Browser (http://promoter.bx.psu.edu/hi-c/view.php). The A/B compartment profiles at 250-kb resolution and the positions of the TAD boundaries were obtained[29].

The contact matrices binned at 10–40-kb resolutions with the positions of the TADs for the IMR90, HMEC, GM12878 (GEO accession: GSE63525)[16], and T47D cells (GEO accession: GSM2827515)[44] were obtained from the 3D Genome Browser[66].

**Analysis of overlap between 4C peaks and genome regions**. We created 4C-Seq bed files at 250-kb resolution ($RPM_{4C} - RPM_{input}$ above the defined threshold) using deepTools (v3.13)[67] and bedtools (v2.27.1)[68]. The files for A/B compartment peaks and those for genes upregulated or not upregulated in LTED cells were compared with the 4C-Seq files, using the mergePeaks tool of Homer (v4.10)[69].

**Chromatin immunoprecipitation (ChIP)-qPCR**. ChIP assays were performed according to the Upstate Biotechnology protocol[70] with modifications. Cells were crosslinked with 1% formaldehyde for 5 min at room temperature. Twenty μL of Dynabeads M-280 Sheep Anti-Mouse IgG (Thermo Fisher Scientific) were bound with 2 μg of a Phospho RNA Polymerase II CTD antibody (Ser5) (MABI0603). Nuclei were isolated and sonicated with a Bioruptor (20 times, 30 s ON/30 s OFF, output; high) (COSMO BIO) to generate DNA fragments. The DNA fragments were incubated overnight at 4 °C with magnetic beads bound with the antibodies. The beads were washed and de-crosslinked for 4 h at 65 °C. RNaseA and Proteinase K were used for RNA and protein digestion, respectively, and the DNA was purified with a QIAquick PCR Purification Kit (Qiagen). DNA enrichment was determined by a qPCR analysis with a Step One Plus system (Applied Biosystems), using SYBR green fluorescence. The input DNA was used to make a standard curve to determine the DNA enrichment. Primer sequences are listed in Supplementary Table 7.

**3C quantitative PCR**. At least $10^7$ MCF7 and LTED cells were crosslinked using 1% formaldehyde in medium for 5 min at room temperature with rotation. Cells were lysed using NP-40 lysis buffer for 15 min at 4 °C. Nuclei were washed and resuspended in restriction buffer and then digested with 1 U/μL HindIII (TaKaRa, 1060AH) overnight at 37 °C while shaking. After inactivating the restriction enzymes by heating at 37 °C for 20 min in the presence of 1.6% sodium dodecyl sulfate (SDS), proximity ligation was done in 7 mL total volume using 350 U T4DNA ligase (TaKaRa, 2011A) for 4 h at 16 °C and then kept for 30 min at room temperature. After de-crosslinking and RNase treatment, the DNA was purified by phenol–chloroform extraction and ethanol precipitation. The 3C template was cleaned using the wizard DNA cleanup system (Promega, A7280). The ligated products (3C template) were assessed using qPCR with the ABI Prism 7300 and Step One Plus system (Applied Biosystems) using TaqMan Fast advanced mastermix (ABI). Primers in this analysis are listed in Supplementary Table 6. For the standard qPCR curve, we used purified BAC DNAs that were digested and ligated. The BAC DNAs containing the target genomic loci were *ESR1*p-BAC for reference and another locus position for a–d (see "BAC clones" section for details).

**Quantitative PCR of genomic DNA for copy number measurement**. Genomic DNA from MCF7 and LTED cells was isolated using a Qiagen DNeasy Kit (Cat. No. 69504), according to the manufacturer's protocol. One column was used for cells grown in 10-cm dish. qPCR was performed with SYBR green fluorescence on an ABI Prism 7300 system (Applied Biosystems). The calibrator was prepared by mixing BAC clones covering tested regions in equimolar amounts (see "BAC clones" section). This mixture was serially diluted to produce a standard curve for normalization against amplification efficiency of each primer set used. Values were further normalized to the genomic DNA of a reference gene, *GAPDH*, before calculating relative fold changes. Primer sets used for analysis are listed in Supplementary Table 6.

**PCR with reverse transcribed RNA**. Total RNA was isolated from cultured cells with TRIzol (Invitrogen). RT was carried out with Rever Tra Ace qPCR RT Master Mix with gDNA Remover (TOYOBO). qPCR was performed with SYBR green fluorescence on an ABI Prism 7300, 7500, and Step One Plus system (Applied Biosystems). Values were normalized to *GAPDH* expression before calculating relative fold changes. Primer sets used for analysis are listed in Supplementary Table 4. For RT–PCR in Fig. 4d, PCR-amplified products were run on agarose gel and stained with SYBR Gold (Thermo Fisher Science). The genomic DNA and the cDNA with and without RT from the extracted RNA were amplified with PCR system on specified number of cycles (Applied Biosystems) using Q5 High-Fidelity DNA Polymerase (NEB). Genomic DNA was used as a control. Primer sets used for analysis are listed in Supplementary Table 8.

**RNA stability assay**. LTED cells were treated with 1 μg/mL actinomycin D (Wako) for 15 min to stop RNA synthesis, and then the residual RNA levels were measured at every 60 min for 300 min. RNA isolation, RT, and PCR analysis were performed as described above. *cMyc* was used as a control for unstable RNA. The primer sets are listed in Supplementary Table 4.

**Immunoblotting**. To prepare the total cell lysate, cells were dissolved in SDS sample buffer containing benzonase (Sigma). Proteins were separated by SDS–polyacrylamide gel electrophoresis and then transferred to a polyvinylidene difluoride membrane, Hybond-ECL (GE Healthcare). The membrane was blocked for 1 h with phosphate-buffered saline (PBS) containing 10% nonfat dry milk and then incubated with primary antibodies against ERα (Santa Cruz Biotechnology, sc-543, 1:1000) and Actin (Sigma, A2103, 1:1000) in Can Get Signal immunostain Solution A (Nacalai Tesque), overnight at 4 °C. The membrane was washed and incubated with secondary antibodies in Can Get Signal immunostain Solution B (Nacalai Tesque) for 60 min at RT. After the membrane was washed, the signals were visualized with Chemi-Lumi One Ultra solution (Nacalai Tesque) and detected by an Amersham Imager 600 (GE Healthcare). The uncropped and unprocessed scans of the blots with molecular marker (MagicMark™ XP, Sigma) are available in Supplementary Fig. 11.

**BAC clones**. BAC DNA used as FISH probes and 3C-qPCR that cover the *Eleanor* TAD (Chr6 q25.1), Chr6 q21 (for FISH), and sites a–d are listed in Supplementary Table 5. BAC DNA was prepared using a large-construct DNA Isolation Kit (Qiagen).

**Fluorescence in situ hybridization**. DNA FISH was performed as follows. Cells grown on coverslips were fixed with 4% formaldehyde and 0.5% Triton X-100 in PBS for 15 min and then permeabilized with 0.5% saponin and 0.5% Triton X-100 in PBS for 20 min. Samples were immersed in 20% glycerol in PBS for 30 min and subjected to four cycles of freeze–thawing by freezing the cells in liquid nitrogen for 30 s each time and then thawing at room temperature. The cells were then treated with 0.1 N HCl for 15 min. For denaturation and hybridization, the cells were incubated in hybridization mixtures (2 × SSC, 50% formamide, 10% dextran sulfate, 1 mg/mL tRNA, and 5–10 μg/mL probe DNA) at 75 °C for 4–10 min (for denaturation of genomic DNA) and then at 37 °C for 48–72 h (for hybridization). For RNA FISH, the same protocol was used, except for the genomic DNA denaturation step. BAC probes were labeled with biotin or digoxigenin in a nick translation mixture (Roche), according to the manufacturer's protocol. After hybridization, the cells were washed with 2 × SSC and 50% formamide at 37 °C for 5 min, followed by 2 × SSC at 37 °C for 5 min. FISH signals were detected with fluorescein isothiocyanate-anti-digoxigenin (Roche) or Cy3-streptavidin (Jackson ImmunoResearch). DNA was counterstained with 4,6-diamidino-2-phenylindole.

**Microscopic and image analysis**. FISH images were obtained using an IX-71 microscope (Olympus) equipped with a 60 × NA1.0 Plan Apo objective lens, a cooled CCD camera (Hamamatsu), and image acquisition software (Lumina Vision Version 2.4; Mitani Corporation). For comparison of the effects of resveratrol, small interfering RNA (siRNA), and LNA treatments, identical image capture conditions were used within a set of experiments. Image stacks of 3D data sets were collected at 0.5–1.0-μm intervals through the z axis, subjected to projections. The signal intensity of RNA FISH was analyzed using a Cellomics CellInsight with HCS studio cell analysis software (Thermo Fisher Scientific).

FISH images in Fig. 2b were obtained with a confocal laser-scanning microscope (LSM 780, Carl Zeiss) equipped with ×63/1.4 Plan-Apochrome objective lens and a cooled CCD camera (Carl Zeiss). Images were acquired using the LSM software (Carl Zeiss).

**Transfection of cells with LNA and siRNA**. Cells were transfected with Antisense LNA GapmeR Negative Control A, LNA *pa-Eleanor(S)-1*, LNA *pa-Eleanor(S)-2* (TaKaRa), siRNA-*GL3*, siRNA-*ESR1* (Nippon EGT), siRNA-*FOXO3* (CST, #6303), and siRNA-*SNX3* (Santa Cruz Biotechnology, sc-41351) using RNAiMAX (Invitrogen). Target sequences for each Antisense LNA and siRNA are listed in Supplementary Table 9. The cells were analyzed 48 or 72 h after transfection for qRT-PCR, RNA FISH, and fluorescence-activated cell sorting (FACS) analyses.

**Transfection of cells**. Cells were transfected with HA-FOXO3a wild type (addgene #1787) using Lipofectamine 2000 (Invitrogen). The cells were monitored 24 h after transfection by qRT-PCR and FACS analyses.

**Cell proliferation assay**. Cell proliferation assay was performed using the Cell Counting Kit-8 (CCK-8; Dojindo). Briefly, cells were seeded in 96-well plates; incubated at 37 °C for 24 h; and transfected with LNA, siRNA, or plasmid. After adding CCK-8 and incubating for 1 h at 37 °C in a humidified $CO_2$ incubator, the degree of yellow color in the medium was measured at an absorbance of 490 nm by a microplate reader (BIO-RAD).

**Apoptosis assay**. Cells were stained using an Annexin-V-FLUOS Staining Kit (Roche), according to the manufacturer's protocol. For FACS analysis, a mixed solution of floating and attached cells were prepared, incubated for 15 min with Annexin V-FLUOS and propidium iodide (PI) labeled solution, and then analyzed with a FACS analyzer (BD FACSCanto™ II and BD FACSCalibur™). For further analysis, we used the FlowJo program (v10.4.1). For microscopic analysis of apoptotic cells, cells grown on a dish were washed with PBS, incubated for 15 min with Annexin V-FLUOS-labeled solution, and then observed under a microscope CKX53 (Olympus) with CellSens standard program (v 1.16). To count cells, merged images were created using ImageJ (v1.49)[71].

**Apoptosis-related gene analysis**. Apoptosis-related genes (Qiagen, https://www.qiagen.com/jp/resources/resourcedetail?id=e5252c51-7513-44a0-b66d-927c53e0eeb2&lang=en) were extracted from the RNA-Seq files[27], and the expression levels were compared in MCF7 and LTED cells.

**Cell vitality assay**. Cell vitality was assessed using VitaBright-48 (VB48, Chemometec), which binds to reduced glutathione in the cell and emits light, using an Automated Cell Analyzer (NucleoCounter NC-250, Chemometec). A mixed solution of floating and attached cells was used for VB48 staining.

**ChIP-Seq and chromatin interaction analysis by paired-end tag sequencing (ChIA-PET) data used in this study**. ChIP-Seq and ChIA-PET data of MCF7 cells (Supplementary Table 10) were obtained from ENCODE Consortium through the UCSC genome browser (University of California, Santa Cruz)[44] and compared with our 4C-Seq data.

**Statistical analysis**. Data in the graphs are mean ± s.e.m. of three independent experiments with triplicated samples. *P* values were assessed using unpaired and two-tailed Student's *t* test (all figures except Figs. 1e, 2c, and 5d, Supplementary Fig. 3e), Mann–Whitney *U* test (for Figs. 1e and 5d), and two-tailed Fisher's exact test (Fig. 2c, Supplementary Fig. 3e), with R. *$P < 0.05$, **$P < 0.01$, ***$P < 0.001$. Values of $P < 0.05$ were considered statistically significant.

**Reporting summary**. Further information on research design is available in the Nature Research Reporting Summary linked to this article.

## Data availability
4C-seq and Hi-C datasets have been deposited in the DNA Data Bank of Japan (DDBJ) Sequence Read Archive with the accession number DRA006154 and DRA007945. RNA-seq datasets used are ones we previously published[27]. Hi-C and ChIP-seq datasets used are publicly available[16,29,44]. Other data that support the findings of this study are available from the corresponding author upon reasonable request. The source data underlying Figs. 1c, e, 2c–e, 3b, c, 4b, c, 5a, b, d, e, and 6a–c and Supplementary Figs. 2a, d, e, 3b–f, 4a, b, d–i, 5c–d, 6a, 7a–g, 8a, c, f, g, 9a, b, and 10b are provided as a Source Data file.

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

## Acknowledgements

We thank all of the members of the Nakao and Saitoh laboratories for discussions and technical assistance. We thank Dr. Gary Felsenfeld (NIH), Dr. Musa Mhlanga (University of Cape Town), and Dr. Yutaka Negishi (University of Cape Town) for critical reading of the manuscript and helpful discussions. We appreciate Dr. Hiroshi Kimura (Tokyo Institute of Technology) for important discussion. We also thank the Advanced Computational Scientific Program of the Research Institute for Information Technology, Kyushu University for providing high-performance computing resources. This work was supported by the JSPS KAKENHI (18H04904, 19H04970 [to K.M.], 18H04802, 18H05527, 19H05244 [to Y.O.], 18H05530 [to I.H.], 18K19479 [to M.N.], and 18H05531, 18K19310 [to N.S.]); JST CREST JPMJCR16G1 [to Y.O.]; Mitsubishi Foundation [to M. N.]; AMED CREST JP16gm0510007 [to M.N.]; The Naito Foundation; Takeda Science Foundation; and The Vehicle Racing Commemorative Foundation [to N.S.]. This work was partly performed in the Cooperative Research Project Program of the Medical Institute of Bioregulation, Kyushu University.

## Author contributions

M.O.A.A. and T.Y. performed most of the experiments. K.M., J.N. and Y.O. performed sequencing and data analyses. H.M., R.P. and I.H. analyzed Hi-C data for TAD profiling. N.S., M.N. and Y.O. conceived, designed, and supervised the work. N.S. and T.Y. wrote the paper with H.N. and M.N. All the authors discussed the results and commented on the manuscript.

## Additional information

**Competing interests:** The authors declare no competing interests.

