## [Peer Review File · Nature Communications]

Reviewers' comments:

Reviewer #1 (Remarks to the Author):

Review for NCOMM-18-19001

Abdalla et al submitted the manuscript "Balanced chromatin interaction for proliferation and death in breast cancer adaptation". Present study is the extension of their previous work on the identification of ncRNA called Eleanors in the paper "a cluster of noncoding RNAs activates the ESR1 locus during breast cancer adaptation". Based on the previous paper published in Nature Communication at 2015, it was assumed that expression of Eleanors are associated with ER expression and localized to the active chromatin domain nearby via chromatin domain regulation. It is experimentally well-conceived that the expressions of Eleanors-TAD genes are co-regulated with ESR1 and long-range interaction is mediated by Eleanors in LTED and LTED-RES cells. However, there are a couple of points to be addressed:

1. Among candidate genes in 6q21, FOXO3 gene is also co-expressed and linked to apoptosis during LTED in Fig. 2. However, after Resveratrol treatment, FOXO3 expression is still high, so it may not be a key regulator for apoptosis in MCF7. In Fig. S3, siFOXO3 was used to demonstrate cell viability during estrogen deprivation. However, the percentage of vital cells are still high (67%), so the differences are not high enough. Authors should use apoptotic assays (Annexin V, as shown for LNA-pa-Eleanor) to clearly demonstrate the critical role of FOXO3 in apoptosis during breast cancer cell adaptation.
2. It is not convincing how Eleanor is linked to FOXO3-mediated apoptosis upon ER deprivation. Considering intimate and paradoxical linkage between ER and FOXO3, as stated in Discussion session, it will be nice to find other genes to explain balance between proliferation and apoptosis in breast cancer cell physiology. Are there any other candidate target genes, direct or indirect, to explain estrogen deprivation adaptation and resveratrol treatment?
3. What is the role of SNX3? It is significantly co-regulated with LTED and LTED-RES in Fig. 2g, but authors did not mention on SNX3 at all. Are there any phenotypic changes after SNX3 knockdown or overexpression?
4. In previous paper published in 2015, authors identified u-Eleanor and demonstrated RNA cloud upon LTED and LTED-RES. In this paper, expression of pa-Eleanor and its switch from the antisense to the sense orientation during LTED were shown in Fig. 3. Since it contains 5'UTR of ESR1 mRNA, it has to be more clearly demonstrate that it is not a part of mRNA. More importantly, how the strand switch occurs has to be understood.
5. Overall, relevance of Eleanor in the regulation of ESR1 and neighboring genes are distinct from long-range interacting genes. Also, the regulation modes by uEleanor and pa-Eleanor are not directly assessed in the manuscript. Considering such ambiguity, the model shown in Fig. 6 is very vague and did not clearly summarize the findings. Model diagram should be more clear.
6. Introduction section is not good well written. General significance of ncRNA-mediated regulation on TAD and its application to cancer cell adaptation should be included to draw more attention. Since the manuscript is the follow-up of previous paper in Nature Communications, introduction should be more interesting. Also, the title is not good representation of the findings.

Reviewer #2 (Remarks to the Author):

The manuscript describes experiments examining the relationship between expression of Eleanor ncRNAs, ESR1, and changes in 3D interactions. The manuscript is potentially interesting but authors need to address the issues raised below, some of which are substantial.

1. Introduction. Authors have a very rudimentary and superficial understanding of 3D organization. Authors should read and cite manuscripts by Rao et al Cell 2014 and 2017 using high resolution Hi-C. These two manuscripts identify small contact domains that probably correspond to what is

generally referred to in the literature as subTADs. Some of these domains correspond to CTCF loops. The rest are referred to by Rao et al Cell 2017 as compartment domains, which remain after depletion of Rad21. Authors should re-write this section of the introduction in the context of these new findings. Based on the findings from these two manuscripts, TADs do not really exist and are probably a computational artefact of low resolution Hi-C data.

2. Page 6, top. "approximately 700 kb chromatin domain". What makes this a domain? Would "region" be more appropriate?

3. Page 3, bottom. "As expected from the nature of "C-technology," massive peaks were detected around the bait site, but sharp transitions occurred at chr6:151,640,001-151,840,001 and 152,680,001-152,880,001, which coincided with TAD boundaries determined by previous Hi-C study using MCF7 cells (dashed blue lines in Fig. 1b)". It is unclear from this description how any of this was calculated. Were the "sharp transitions" in 4C signal calculated by an objective computational approach or just visually? It appears that TAD boundaries were calculated by binning the Hi-C data into 250 kb bins. This would position the boundaries at a resolution of +/- 250 kb, which represents a very large region when making the arguments the authors want to state. An objective observer, asked to place the boundaries of the 4C-seq, would not place them where the authors have. The region where the 4C-seq becomes background is not at the borders the authors have marked as the TAD boundaries, but further outside. Alternatively, one could argue that based on visual inspection the 4C-seq suggests the existence of three domains within the region claimed to be a TAD.

4. Page 7, top. "To investigate the significance of the Eleanor TAD, we measured the transcriptional activities of genes both within and outside the Eleanor TAD". To emphasize the point that the location of the TAD boundaries the authors have chosen is arbitrary and it depends on the resolution of the Hi-C data, I'm enclosing a picture of the same region using Hi-C data from Rao et al in GM12878 cells containing 4.5 billion reads instead of a couple hundred. The differences are obvious, although one could argue that this is due to the different cell type. However, the same region looks very similar in HCT116 cells at the same resolution. Is the ESR1 gene expressed in GM12878 cells? The domains the authors claim to be a TAD looks considerably different, with several CTCF loops that presumably would interfere with enhancer-promoter interactions and with the co-regulation the authors claim takes place within this domain. Are these CTCF sites present in MCF7 cells? If so, they presumably would form the same CTCF loops. Where are the different co-regulated genes present with respect to these loops? Are they all inside the loops or are some outside?

5. Page 7, bottom. "Upon further analysis of our 4C-Seq data, we observed intra-chromosomal ESR1 promoter interactions along the entirety of chromosome 6 in MCF7 and LTED cells". It is surprising that authors observe such long-range interactions in the 4C data. Based on the description in the Methods, it appears that authors only subtracted the input from the experimental with no distance normalization. Authors should use a standard 4C analysis pipeline such as that described by de Laat and collaborators in PMID 22961246.

6. Page 8, top. "However, there was a tendency for the 4C peaks to coincide with A compartments throughout the chromosome, suggesting that the ESR1 promoter region in the Eleanor TAD interacted with genes in the A compartments of multiple regions of chromosome 6. Consistently, ESR1 promoter interacting regions showed moderate coincidence with genes that were upregulated in LTED cells (red bars below 4C-Seq profiles in Fig. 2a)". Please explain in quantitative terms what "tendency" and "moderate" means. If the authors want to make the argument that regions that interact with the ESR1 gene represent upregulated genes, it may be better to plot the RNA-seq data. Also, it would help the reader assess the overlap between upregulated genes and 4C interactions if regions of interest are shown at a higher resolution, perhaps in the supplemental figures.

7. Page 9, bottom. "we inhibited Eleanors with resveratrol in LTED cells and performed 3C-qPCR analysis. We found that the ESR1 promoter and the b site were dissociated after resveratrol treatment (Fig. 2f). It should be noted that the transcriptional levels of the b site genes were mostly unchanged". It seems from Figure 2G that transcription of SNX3 and LACE1 changed quite a bit. How long was the resveratrol treatment? Is it possible that the RNAs for the other genes are more stable and that's why change was not detected?

8. Page 10, top. "Taken together, these data show that the chromatin interaction between ESR1 and FOXO3 loci requires Eleanors, and that the interaction plays a role in coregulation of genes in the Eleanor TAD in LTED cells. Eleanor transcription is required to resist cell death in the transition from MCF7 to LTED and that the FOXO3 containing TAD somehow is permissive for Eleanor transcription". Perhaps I'm missing something related to this conclusion. My understanding is that the authors observe correlations among various events with no basis for casual inferences. First, does treatment with resveratrol affect Eleanors and this then affects transcription of coding genes? Or are the two events correlated without an established causal relationship? Second, the authors never show causality between interactions and transcription. Authors assume that interactions determine transcription, but it is equally possible, and perhaps more likely, that changes in transcription are determined by binding of protein complexes to the promoters and regulatory regions of genes, and that these proteins then mediate the interactions. Loss of interactions after resveratrol treatment could be a consequence of decreased transcription, which the authors argue it does not change (see #7), but Figure 2G appears to suggest that it does. Does RNA pol II remain unchanged after treatment? The TAD is not permissive for transcription, unless the authors are talking about specific CTCF loops, which do not necessarily correspond to TADs.

9. Page 11, top. "because most other coding (exons) and non-coding regions in the Eleanor TAD were transcribed in the same direction". What does "most" mean, 51% or 99%? Please be quantitative.

10. Page 11, bottom. "knockdown of ESR1 mRNA with RNAi or specific degradation of ER protein with a chemical compound ICI 782,780, had no effect on pa-Eleanor(S) expression". Evidence showing that ESR1 protein was depleted in these experiments should be shown.

11. Page 12, top. "To investigate the molecular function of pa-Eleanor(S), we knocked it down in LTED cells using Antisense LNA Gapmer (LNA) (Fig. 4a)". It is surprising that KD of pa-Eleanor had a modest effect when measured by qPCR (Figure 4a) but a very strong effect when measured by FISH (Figure 4C). Please comment on this discrepancy.

12. Page 12, bottom. "To investigate these, we first knocked down ESR1 mRNA in LTED cells and found that the transcriptional levels of genes in the Eleanor TAD, including pa-Eleanor(S) and the Eleanor RNA cloud, did not change (Supplementary Fig. 5a-c). Therefore, Eleanors are specific ncRNAs that play a role in the activation of the chromatin domain. Consistent with this result, ICI 182,780 did not change the expression levels of either Eleanor TAD genes or pa-Eleanor(S)". Authors should show that the levels of ESR1 protein actually changed in these experiments. Changes in ESR1 RNA levels shown in supplemental Figure 5A are very small.

13. Page 13, bottom. "To investigate the function of pa-Eleanor(S) in gene regulation through chromatin structures, we performed 3C-qPCR analysis. We found that the interaction between the ESR1 promoter and the b site at 6q21 reduced with pa-Eleanor(S) knockdown in LTED cells (Fig. 5a), supporting the idea that the ncRNA was involved in the long-range chromatin interaction. In contrast, transcriptional levels of the genes at 6q21 showed little to-no change in the LTED cells with pa-Eleanor(S) knockdown (Fig. 5b)". This is a critical experiment and conclusion that should not rely on just one 3C experiment. Authors should perform 4C and examine all interactions between ESR1 and other sequences in the chromosome. Furthermore, gene expression changed

significantly in at least one of the two experiments shown in Figure 5B. These changes in expression could account for the changes in interactions shown in Figure 5A.

Reviewer #3 (Remarks to the Author):

In this manuscript, the authors investigated the long-range chromatin TADs interaction and their consequence on balance of LTED MCF-7 cells growth and apoptosis. They found lncRNA pa-Eleanor (S) mediated the interaction of two TADs. Silence of pa-Eleanor inhibited the formation of Eleanor clouds, the binding of the b site of 6q21 with ESR1 promoter and promoted LTED cell apoptosis. And inhibition of pa-Eleanor mimics the effect of resveratrol treatment. In general, this manuscript is the extension of the previous study (Nat Commun 6, 6966 (2015)) and clarified the mechanism of LTED cells primed for apoptosis. However, the roles of 6q21 TAD in promoting co-upregulation of genes in the Eleanor TAD were still unknown. Some issues need to be addressed.

Major comments:

1. This manuscript investigated the balanced chromosomal interaction between genes for cell proliferation (ESR1) and death (FOXO3) in LTED cells, mediated by Eleanors. However, gene expression in Eleanors TAD didn't affect the activation of FOXO3 contained 6q21 TAD. Also, there were no direct evidence to prove the interaction between the b site of 6q21 with ESR1 promoter promoted the transcription of Eleanors and ESR1. It seems only pa-Eleanor regulated ESR1 expression determined LETD cell growth or apoptosis. The authors should provide more experiments to show the functional interaction of 6q21 on ESR1 promoter.
2. The group's previous paper found u-Eleanor is involved in transcriptional activation of the ESR1 locus and maintains downstream intragenic Eleanor in LTED cells. The pa-Eleanor reported in the present study displayed the similar functions. The authors should discuss the relationship between u-Eleanor and pa-Eleanor. And was it a general mechanism for Eleanor ncRNAs in regulating ESR1 expression?

Minor comments:

1. For all the cell apoptosis experiments, the authors had better provide flow cytometry results stained by annexin V/PI.
2. For all the flow cytometry experiments showed in this manuscript, the authors should also provide the isotype-treatment curve for each group.
3. In figure 5a, there was interaction between c site with ESR1 promoter, which was not consistent with fig 2d, 2f. the authors should explain the results.
4. In sup fig5e, the basal cell viability was largely different. The experimental conditions should be optimized.

Reviewer #1 (Remarks to the Author):

Review for NCOMM-18-19001

Abdalla et al submitted the manuscript “Balanced chromatin interaction for proliferation and death in breast cancer adaptation”. Present study is the extension of their previous work on the identification of ncRNA called Eleanors in the paper “a cluster of noncoding RNAs activates the ESRI locus during breast cancer adaptation”. Based on the previous paper published in Nature Communication at 2015, it was assumed that expression of Eleanors are associated with ER expression and localized to the active chromatin domain nearby via chromatin domain regulation. It is experimentally well-conceived that the expressions of Eleanors-TAD genes are co-regulated with ESRI and long-range interaction is mediated by Eleanors in LTED and LTED-RES cells. However, there are a couple of points to be addressed:

1. Among candidate genes in 6q21, FOXO3 gene is also co-expressed and linked to apoptosis during LTED in Fig. 2. However, after Resveratrol treatment, FOXO3 expression is still high, so it may not be a key regulator for apoptosis in MCF7. In Fig. S3, siFOXO3 was used to demonstrate cell viability during estrogen deprivation. However, the percentage of vital cells are still high (67%), so the differences are not high enough. Authors should use apoptotic assays (Annexin V, as shown for LNA-pa-Eleanor) to clearly demonstrate the critical role of FOXO3 in apoptosis during breast cancer cell adaptation.

We thank the reviewer for this important comment, because it made us aware that our previous description was insufficient, and therefore we have revised the manuscript. As pointed out, *FOXO3* expression is up-regulated in LTED cells and it remains high after resveratrol treatment. This is one of the reasons why we propose that it function as a key regulator for apoptosis in LTED-RES cells. Please note that we do not claim that *FOXO3* may be a key regulator for apoptosis in MCF7 cells. *FOXO3* is a tumor suppressor and promotes apoptosis (Ekoff et al., 2007; Skurk et al., 2004). We have now stated this clearly in the text on page 11, line 5.

We agree that the percentage of vital cells is already high (67%) in the control knock down cells (si*GL3*), so the effect of si*FOXO3* may not have been high enough in the previous **Supplementary Fig. 3**. Therefore, according to this reviewer’s suggestion, we performed the apoptosis assay with Annexin V to clarify the role of *FOXO3* in apoptosis. The results showed that si*FOXO3* reduced apoptosis in LTED (2M) cells (new **Supplementary Fig. 4d**). The effect of si*FOXO3* was statistically significant, as $P= 1.82E-05$, and more prominent in LTED (2M) than in MCF7 cells, as expected. Our additional CCK-8 assay also supported the fact that the *FOXO3*

knockdown rescued the proliferation of LTED (2M) cells (new **Supplementary Fig. 4b**). For further confirmation, we overexpressed *FOXO3* in LTED cells, and found that it induced cell death (new **Supplementary Fig. 4e-g**). Altogether, these results clearly demonstrate that *FOXO3* functions to induce apoptosis in breast cancer cell adaptation under estrogen deprivation. We presented these new data in the new **Supplementary Fig. 4b, d-g**, and described them in the revised manuscript on page 10, line 2.

2. It is not convincing how Eleanor is linked to FOXO3-mediated apoptosis upon ER deprivation. Considering intimate and paradoxical linkage between ER and FOXO3, as stated in Discussion session, it will be nice to find other genes to explain balance between proliferation and apoptosis in breast cancer cell physiology. Are there any other candidate target genes, direct or indirect, to explain estrogen deprivation adaptation and resveratrol treatment?

To address this reviewer's comment, we re-analyzed our previously published RNA-Seq data and extracted 16 apoptotic genes that are co-regulated with *FOXO3* in LTED and LTED-RES cells (new **Supplementary Fig. 6b**, new **Supplementary Table 3**). They are *JUN* (c-JUN), *CDKN1A* (P21 / WAF1), *FOXO1* and others (new **Supplementary Fig. 6b**). Among them, three genes are encoded on chromosome 6, and all of them are located in the A-compartment and therefore may be co-upregulated with the *Eleanor* TAD in LTED cells (new **Supplementary Table 3**). The upregulation of multiple apoptotic genes may explain the apoptotic prone-nature of LTED cells, and apoptosis can be triggered by inhibition of *Eleanors*. These data are presented in the new **Supplementary Fig. 6b** and **Supplementary Table 3**. We discuss these results in the revised text on page 11, line 13.

3. What is the role of SNX3? It is significantly co-regulated with LTED and LTED-RES in Fig. 2g, but authors did not mention on SNX3 at all. Are there any phenotypic changes after SNX3 knockdown or overexpression?

As the reviewer pointed out, *SNX3* expression was co-regulated with *ESR1* in LTED and LTED-RES cells. The *SNX3* protein is involved in intracellular trafficking (Chioy et al., 2012; Haft et al., 1998). To understand its function in endocrine therapy resistance in breast cancer, we knocked down *SNX3*, and found that it had no effect on cell proliferation. We show these data in the new **Supplementary Fig. 4h, i**, and describe them in the text on page 10, line 8.

4. In previous paper published in 2015, authors identified u-Eleanor and demonstrated RNA cloud upon LTED and LTED-RES. In this paper, expression of pa-Eleanor and its switch from the

antisense to the sense orientation during LTED were shown in Fig. 3. Since it contains 5'UTR of ESR1 mRNA, it has to be more clearly demonstrate that it is not a part of mRNA.

In the revised text, we show the appropriate position of the TSS for the *ESR1* mRNA variant 1 (new **Fig. 4a**), which starts downstream of *pa-Eleanor (S)* and is produced mostly in mammary cells (Consortium, 2012; Gannon et al., 2001; Reid et al., 2002). The 5'UTR of this *ESR1* mRNA is not included in *pa-Eleanor(S)*. For confirmation, we performed an RT-PCR analysis and did not detect a transcript that is contiguous from *pa-Eleanor(S)* to exon 1 of the *ESR1* mRNA (new **Fig. 4d**, lane 7). In contrast, we successfully detected the transcript from upstream of exon 1 of the *ESR1* mRNA, corresponding to *pa-Eleanor* (new **Fig. 4d**, lane 3), as well as the one from exon 1 of the *ESR1* mRNA (new **Fig. 4d**, lane 11). These results indicate that *pa-Eleanor(S)* is not contiguous with the *ESR1* mRNA, and are now stated more clearly in the revised text on page 12, lines 11 and 18, and page 43, line 22-page 44, line 4.

In addition, we found that the cellular phenotype was different in each knockdown. With the *pa-Eleanor(S)* knockdown, the genes in the *Eleanor* TAD and the *Eleanor* cloud formation were repressed (new **Fig. 5a-c**), while they were not affected with the *ESR1* mRNA knockdown (new **Supplementary Fig. 8a-d**). Altogether, we concluded that *pa-Eleanor* is independent of the mRNA.

More importantly, how the strand switch occurs has to be understood.

We agree that the mechanism for the transcriptional switching is important. It is possible that RNA polymerase II transcription in the opposite direction may interfere with the movement of RNA polymerase II toward the *ESR1* gene in MCF7 cells, which could be relieved in LTED cells. This idea is shared with the transcriptional interference in yeast and other eukaryotes (Hao et al., 2016; Mazo et al., 2007), and is now described in the discussion section of the revised manuscript on page 18, line 12.

5. Overall, relevance of Eleanor in the regulation of ESR1 and neighboring genes are distinct from long-range interacting genes. Also, the regulation modes by uEleanor and pa-Eleanor are not directly assessed in the manuscript. Considering such ambiguity, the model shown in Fig. 6 is very vague and did not clearly summarize the findings. Model diagram should be more clear.

According to this reviewer, we have revised the model diagram in the new **Fig. 7** for clarity. We also made it clear that the relevance of *Eleanor* in the regulation of *ESR1* and its neighboring genes is distinct from that in long-range interacting genes, and we have revised the discussion section of the text on page 16, line 13.

To understand the regulation modes of *pa-Eleanor* and *u-Eleanor*, we investigated the expression of *pa-Eleanor* and *u-Eleanor* upon the knockdown of each ncRNA. The *u-Eleanor* knockdown did not change the expression of *pa-Eleanor* (new **Supplementary Fig. 7f**). In contrast, the knockdown of *pa-Eleanor(S)* suppressed the expression level of *u-Eleanor* (new **Supplementary Fig. 7g**). These results suggest that *pa-Eleanor* upregulates the *ESR1* transcription through *u-Eleanor*, or may directly affect the expression of *ESR1*. These results are summarized in the new **Supplementary Fig. 7h**, and described in the revised text on page 14, line 4.

6. Introduction section is not good well written. General significance of ncRNA-mediated regulation on TAD and its application to cancer cell adaptation should be included to draw more attention. Since the manuscript is the follow-up of previous paper in Nature Communications, introduction should be more interesting. Also, the title is not good representation of the findings.

According to the reviewer, we created a new paragraph stating the significance of the ncRNA-mediated regulation on TAD and its possible application to cancer cell adaptation on page 4, line 9 of the revised manuscript.

We also changed the title to “*Eleanors* activate the chromatin domain, with balancing cell death and proliferation in breast cancer adaptation”

Reviewer #2 (Remarks to the Author):

The manuscript describes experiments examining the relationship between expression of Eleanor ncRNAs, ESR1, and changes in 3D interactions. The manuscript is potentially interesting but authors need to address the issues raised below, some of which are substantial.

1. Introduction. Authors have a very rudimentary and superficial understanding of 3D organization. Authors should read and cite manuscripts by Rao et al Cell 2014 and 2017 using high resolution Hi-C. These two manuscripts identify small contact domains that probably correspond to what is generally referred to in the literature as subTADs. Some of these domains correspond to CTCF loops. The rest are referred to by Rao et al Cell 2017 as compartment domains, which remain after depletion of Rad21. Authors should re-write this section of the introduction in the context of these new findings. Based on the findings from these two manuscripts, TADs do not really exist and are probably a computational artefact of low resolution Hi-C data.

We thank this reviewer for providing these important comments. We agree that the two publications (Rao et al., 2014; Rao et al., 2017) are very important, in which high resolution Hi-C was performed and exhibited contact matrices binned at 1-kb resolution, and novel principles of chromatin looping. The reports identified loop domains that demarcate contact domains, which may correspond to those generally referred to as sub-TADs. In addition, recent reports employing single cell Hi-C analyses indicated the variabilities of TADs among cells (Bintu et al., 2018; Finn et al., 2019). Accordingly, we have rewritten our introduction in the context of these new findings on page 3, line 18.

As described below, we would like to clarify that we had analyzed the *Eleanor* TAD at a 40-kb resolution (**Fig. 1b**), in contrast to the 250-kb or lower resolution in the previous report (Lieberman-Aiden et al., 2009). In this revised manuscript, we further analyzed several Hi-C data, including those from the work by Rao et al. (Rao et al., 2014), and consistently detected the *Eleanor* TAD (new **Supplementary Fig. 2c**). Considering this and many other reports regarding TADs (Donaldson-Collier et al., 2019; Le Dily et al., 2019; Nuebler et al., 2018; Schwarzer et al., 2017; Szabo et al., 2018), it is beyond the scope of this manuscript to discuss whether TADs do not really exist or not.

2. Page 6, top. “approximately 700 kb chromatin domain”.

What makes this a domain? Would “region” be more appropriate?

We changed the sentence to “approximately 700 kb chromatin region” on page 6, line 5.

3. Page 3, bottom. “As expected from the nature of “C-technology,” massive peaks were detected around the bait site, but sharp transitions occurred at chr6:151,640,001-151,840,001 and 152,680,001-152,880,001, which coincided with TAD boundaries determined by previous Hi-C study using MCF7 cells (dashed blue lines in Fig. 1b)”.

It is unclear from this description how any of this was calculated. Were the “sharp transitions” in 4C signal calculated by an objective computational approach or just visually?

We again thank the reviewer for raising this important point. Unlike the directionality index or insulation square analysis for the Hi-C analysis, the positioning of the border in 4C-Seq is not well established. In the previous manuscript, we determined the 4C-Seq boundaries by visually inspection. With this reviewer’s comment, we made a large effort to determine them quantitatively. We applied the Tukey-Kramer method to the sequencing read numbers in the region of **151,000,000-153,000,000**, with 100-kb resolution (new **Supplementary Fig. 2a**). In the 4C-Seq of MCF7 and LTED cells, we found that the regions of approximately **151,700,000** and **152,700,000** were statistically distinct from the neighboring regions ($P < 0.05$), suggesting that they are the places

where the sharp transitions occur. This observation was consistent among all four of the experiments using MCF7 and LTED cells (new **Supplementary Fig. 2a**). We found that the region inside the 4C-Seq, at approximately 151,900,000 and 152,200,000, might be distinct from the neighboring region, which suggests the possible presence of three sub-domains, but they were observed inconsistently (new **Supplementary Fig. 2a**). Due to this inconsistency, we did not further pursue the sub-TAD.

It appears that TAD boundaries were calculated by binning the Hi-C data into 250 kb bins. This would position the boundaries at a resolution of +/- 250 kb, which represents a very large region when making the arguments the authors want to state. An objective observer, asked to place the boundaries of the 4C-seq, would not place them where the authors have. The region where the 4C-seq becomes background is not at the borders the authors have marked as the TAD boundaries, but further outside. Alternatively, one could argue that based on visual inspection the 4C-seq suggests the existence of three domains within the region claimed to be a TAD.

We deeply apologize for the confusing sentence in the previous manuscript, “Positions of TAD boundaries and A/B compartments (250-kb bins) in MCF7 cell were also derived from the same study”. It should have been the “positions of the TAD boundaries were determined by insulation square analysis using 40-kb bins, while the A/B compartment profiles were calculated at 250-kb resolutions (Barutcu et al., 2015)”. Therefore, we have more precisely rewritten the method section of the manuscript on page 25, line 14.

In the revised manuscript, we also analyzed the Hi-C data of IMR90 (human fetal lung cells), HMEC (human mammary epithelial cells) (Rao et al., 2014), and T47D cells (human ER positive breast cancer cells) (Consortium, 2012), and the high resolution analyses revealed the contact domains or sub-TADs more clearly. The directional index revealed that the same regions form boundaries (new **Supplementary Fig. 2b**), regardless of whether they were analyzed with higher resolutions (10-kb for HMEC and IMR90, 40-kb for T47D, and 10-kb for GM12878) than that for MCF7 cells (Barutcu et al., 2015) (**Fig. 1b**). All showed the TAD boundaries at **151,700,000** and **152,700,000**, and thus matched the 4C-Seq boundaries that we described above.

In conclusion, we have determined the 4C-Seq boundaries quantitatively and found that they matched the TAD boundaries consistently determined with the four different cell types. These findings are described in the results section on page 6, line 17.

4. Page 7, top. “To investigate the significance of the Eleanor TAD, we measured the transcriptional activities of genes both within and outside the Eleanor TAD”.

To emphasize the point that the location of the TAD boundaries the authors have chosen is arbitrary and it depends on the resolution of the Hi-C data, I'm enclosing a picture of the same region using Hi-C data from Rao et al in GM12878 cells containing 4.5 billion reads instead of a couple hundred. The differences are obvious, although one could argue that this is due to the different cell type. However, the same region looks very similar in HCT116 cells at the same resolution. Is the ESRI gene expressed in GM12878 cells?

We thank the reviewer for providing the Hi-C data. According to the comment, we analyzed the TAD boundaries at 40-kb bins for these Hi-C data of GM12878 cells (Rao et al., 2014) on the 3D Genome Browser. Alignments of the TAD profiles of the GM12878 (containing 4.5 billion reads) and MCF7 (0.15 billion reads) cells (Barutcu et al., 2015) showed that the TAD in GM12878 and the one in MCF7 cells (0.15 billion reads) shared most of the region (151,825,000-153,000,000 for GM12878 cells, and 151,750,000-152,750,000 for MCF7 cells). As the reviewer pointed out, the *ESRI* gene is not expressed in both GM12878 and HCT116 cells (RNA-Seq of ENCODE). These data are presented in the new **Supplementary Fig. 2c**, and described in the revised manuscript on page 6, line 20.

The domains the authors claim to be a TAD looks considerably different, with several CTCF loops that presumably would interfere with enhancer-promoter interactions and with the co-regulation the authors claim takes place within this domain. Are these CTCF sites present in MCF7 cells? If so, they presumably would form the same CTCF loops. Where are the different co-regulated genes present with respect to these loops? Are they all inside the loops or are some outside?

According to the reviewer, we now show the data for the CTCF ChIA-PET of MCF7 cells (Consortium, 2012) at the bottom of the new **Supplementary Fig. 2c**. These loops that were anchored with CTCF may well correspond to the CTCF loops that were described in Rao et al. (Rao et al., 2014). There are no CTCF loops that traverse beyond the *Eleanor* TAD. Please note that no enhancer elements governing the co-regulation of the genes in the *Eleanor* TAD have been described. Therefore, it may not be appropriate to speculate whether the CTCF loops would interfere with the enhancer-promoter interactions, interfere with the co-regulation of the genes within this domain, or represent local promoter-enhancer interactions.

5. Page 7, bottom. "Upon further analysis of our 4C-Seq data, we observed intra-chromosomal ESRI promoter interactions along the entirety of chromosome 6 in MCF7 and LTED cells". It is surprising that authors observe such long-range interactions in the 4C data. Based on the description in the Methods, it appears that authors only subtracted the input from the experimental

with no distance normalization. Authors should use a standard 4C analysis pipeline such as that described by de Laat and collaborators in PMID 22961246.

According to this reviewer, we re-analyzed our 4C-Seq data using the suggested standard pipeline described by de Laat and colleagues (van de Werken et al., 2012). In this method, the interactions were normalized against the distance from the bait. We again found interactions between the *ESR1* promoter and 6q21 in LTED cells. These new data are presented in the new **Supplementary Fig. 3a**, labeled as 4C-Seq (normalized against distance).

6. Page 8, top. “However, there was a tendency for the 4C peaks to coincide with A compartments throughout the chromosome, suggesting that the ESR1 promoter region in the Eleanor TAD interacted with genes in the A compartments of multiple regions of chromosome 6. Consistently, ESR1 promoter interacting regions showed moderate coincidence with genes that were upregulated in LTED cells (red bars below 4C-Seq profiles in Fig. 2a)”.

Please explain in quantitative terms what “tendency” and “moderate” means.

We apologize for not describing the correlations in quantitative terms. We analyzed the overlap between the 4C peaks and the A compartments along chromosome 6 with a 250-kb bin resolution. In LTED and MCF7 cells, more than 60% of the regions showed the overlap. Similarly, we found that 60% of the regions showed the overlap between the 4C peaks and the genes upregulated in LTED cells. We show these new data in the new **Supplementary Fig. 3b, c**, and describe them in the revised manuscript on pages 8, lines 10 and 17.

If the authors want to make the argument that regions that interact with the ESR1 gene represent upregulated genes, it may be better to plot the RNA-seq data. Also, it would help the reader assess the overlap between upregulated genes and 4C interactions if regions of interest are shown at a higher resolution, perhaps in the supplemental figures.

We have plotted the genes upregulated in LTED cells as red bars below the A/B compartment graph in the new **Fig. 2a** and the new **Supplementary Fig. 3a**. The bars are derived from our previous RNA-Seq data (Tomita et al., 2015). The data suggested that the regions that interact with the *ESR1* gene correlated with the genes upregulated in LTED cells. We also show an enlarged view in the new **Supplementary Fig. 3a**, to help the readers assess the overlap between upregulated genes and 4C interactions. Furthermore, we calculated the overlap between upregulated genes and 4C interactions. The results are presented in the new **Supplementary Fig. 3c.**, and described in the text on page 8, line 17.

7. Page 9, bottom. “we inhibited *Eleanors* with resveratrol in LTED cells and performed 3C-qPCR analysis. We found that the *ESR1* promoter and the *b* site were dissociated after resveratrol treatment (Fig. 2f).

*It should be noted that the transcriptional levels of the b site genes were mostly unchanged”. It seems from Figure 2G that transcription of *SNX3* and *LACE1* changed quite a bit. How long was the resveratrol treatment? Is it possible that the RNAs for the other genes are more stable and that’s why change was not detected?*

To address the comments, first, we revised the text to describe the result, as follows. “It should be noted that the *FOXO3* gene remained upregulated in LTED-RES (new **Fig. 3c** and new **Supplementary Fig. 5c**)” on page 11, line 5. “The expression of other genes in 6q21, *SNX* and *LACE1*, was reduced (new **Fig. 3c**, new **Supplementary Fig. 5c** and **Supplementary Table 2**)” on page 11, line 7.

Next, we address the specific question. We treated the LTED cells with resveratrol for 24 hours.

Finally, to address the point raised by this reviewer, we measured the stabilities of RNAs for *SNX3*, *LACE1*, *FOXO3* and others. We found that the stability of the *FOXO3* RNA was lower than those of the *LACE1* and *SNX3* RNAs. Therefore, it is unlikely that the maintenance of *FOXO3* transcription was due to its stability. These new results are presented in the new **Supplementary Fig. 6a**, and described in the revised manuscript on page 11, line 9.

8. Page 10, top. “Taken together, these data show that the chromatin interaction between *ESR1* and *FOXO3* loci requires *Eleanors*, and that the interaction plays a role in coregulation of genes in the *Eleanor* TAD in LTED cells. *Eleanor* transcription is required to resist cell death in the transition from MCF7 to LTED and that the *FOXO3* containing TAD somehow is permissive for *Eleanor* transcription”.

*Perhaps I’m missing something related to this conclusion. My understanding is that the authors observe correlations among various events with no basis for casual inferences. First, does treatment with resveratrol affect *Eleanors* and this then affects transcription of coding genes? Or are the two events correlated without an established causal relationship?*

We thank the reviewer for pointing out the need for this clarification. We would like to make sure that we have shown that the resveratrol treatment affected *Eleanors*, but did not affect the transcription of the coding genes in 6q21. We demonstrated these data in the previous **Fig. 2g** (new **Fig. 3c**). We have also described, “More importantly, the activation of the *FOXO3* gene in the 6q21

domain is autonomous and not dependent on a long-range chromosomal interaction via *Eleanors*.”
on page 16, line 2.

Second, the authors never show causality between interactions and transcription. Authors assume that interactions determine transcription, but it is equally possible, and perhaps more likely, that changes in transcription are determined by binding of protein complexes to the promoters and regulatory regions of genes, and that these proteins then mediate the interactions. Loss of interactions after resveratrol treatment could be a consequence of decreased transcription, which the authors argue it does not change (see #7), but Figure 2G appears to suggest that it does.

We agree with the reviewer that it is important to mention the causality between interaction and transcription. We also agree that it is likely that the loss of interactions after the resveratrol treatment could be a consequence of the decreased transcription of the *Eleanor* TAD. We agree that once transcription has been activated by *Eleanors*, many transcription factors are gathered together, and then the long-range chromatin interaction may be established. Previous reports also showed that the enhancer region of *Bcl11b* (an essential gene for early T cell differentiation) was transcribed and then the chromatin interaction changed (Isoda et al., 2017). Therefore, we have revised the discussion section on page 16, line 21.

Does RNA pol II remain unchanged after treatment? The TAD is not permissive for transcription, unless the authors are talking about specific CTCF loops, which do not necessarily correspond to TADs.

According to the reviewer, we performed ChIP-qPCR, and found that the RNA polymerase II binding to the TSS region of the genes in the *Eleanor* TAD was reduced by the resveratrol treatment of LTED cells (new **Supplementary Fig. 2e**). In contrast, RNA polymerase II binding to the TSSs of *SNX3* and *FOXO3* in 6q21 did not change (new **Supplementary Fig. 5d**). Both are consistent with each gene's activity. We describe these findings in the revised manuscript on page 7, line 16 and page 11, line 10.

9. Page 11, top. “because most other coding (exons) and non-coding regions in the Eleanor TAD were transcribed in the same direction”.

What does “most” mean, 51% or 99%? Please be quantitative.

We agree that a quantitative description is important. We found that most (90%) of the *ESR1* region (chr6:152,128,814-152,424,408) was transcribed in the sense direction, based on a re-analysis of our

stranded RNA-Seq data in MCF7 and LTED cells (Tomita et al., 2015), as shown in the graph below. We now describe these data on page 12, line 11.

Sequencing reads mapped at the *ESR1* gene. The numbers corresponding to sense and antisense RNA were counted and normalized against the total number of mapped reads.

10. Page 11, bottom. “knockdown of ESR1 mRNA with RNAi or specific degradation of ER protein with a chemical compound ICI 782,780, had no effect on pa-Eleanor(S) expression”. Evidence showing that ESR1 protein was depleted in these experiments should be shown.

According to the reviewer, we performed immunoblotting and the results showed that the ER protein levels were decreased by si*ESR1* and ICI 182,780 treatments in LTED cells. These new data are now presented in the new **Supplementary Fig 8b, e**, and described in the revised manuscript on page 13, line 1.

11. Page 12, top. “To investigate the molecular function of pa-Eleanor(S), we knocked it down in LTED cells using Antisense LNA GapmeR (LNA) (Fig. 4a)”. It is surprising that KD of pa-Eleanor had a modest effect when measured by qPCR (Figure 4a) but a very strong effect when measured by FISH (Figure 4C). Please comment on this discrepancy.

We agree with the reviewer. This is consistent with our previous observation of the large effect of the *u-Eleanor* knockdown (Tomita et al., 2015). The result suggests that the level of *pa-Eleanor* may be critically regulated and a subtle reduction leads to large molecular and cellular alterations, including the *Eleanor* cloud disappearance (new **Fig. 5c-e**). We discuss this discrepancy in the revised text on page 13, line 20.

12. Page 12, bottom. “To investigate these, we first knocked down ESRI mRNA in LTED cells and found that the transcriptional levels of genes in the Eleanor TAD, including pa-Eleanor(S) and the Eleanor RNA cloud, did not change (Supplementary Fig. 5a-c). Therefore, Eleanors are specific ncRNAs that play a role in the activation of the chromatin domain. Consistent with this result, ICI 182,780 did not change the expression levels of either Eleanor TAD genes or pa-Eleanor(S)”. Authors should show that the levels of ESRI protein actually changed in these experiments. Changes in ESRI RNA levels shown in supplemental Figure 5A are very small.

According to the reviewer’s comment, we performed immunoblotting, which revealed that the ER protein level was clearly decreased by siESRI and ICI 182,780 treatments in LTED cells. These new data are shown in the new **Supplementary Fig. 8b, e**, and described on page 14, line 14.

13. Page 13, bottom. “To investigate the function of pa-Eleanor(S) in gene regulation through chromatin structures, we performed 3C-qPCR analysis. We found that the interaction between the ESRI promoter and the b site at 6q21 reduced with pa-Eleanor(S) knockdown in LTED cells (Fig. 5a), supporting the idea that the ncRNA was involved in the long-range chromatin interaction. In contrast, transcriptional levels of the genes at 6q21 showed little to-no change in the LTED cells with pa-Eleanor(S) knockdown (Fig. 5b)”.

This is a critical experiment and conclusion that should not rely on just one 3C experiment. Authors should perform 4C and examine all interactions between ESRI and other sequences in the chromosome.

According to the reviewer’s suggestion, we performed Hi-C, which is equivalent to 4C, but more suitable for a genome-wide analysis. We found that all interactions between ESRI and other sequences in the chromosome did not dramatically change by the Eleanor inhibition with resveratrol (new **Fig. 3a** and new **Supplementary Fig. 5a**). However, a detailed examination suggested that the contact between the ESRI promoter and FOXO3 became unstable with resveratrol treatment (new **Fig. 3a** and new **Supplementary Fig. 5a, b**). We present these data in the new **Fig. 3a**, the new **Supplementary Fig. 5a, b**, and describe them in the revised text on page 10, line 16.

Furthermore, gene expression changed significantly in at least one of the two experiments shown in Figure 5B. These changes in expression could account for the changes in interactions shown in Figure 5A.

We thank the reviewer for careful inspection and providing the interesting possibility. As the reviewer mentioned, the change in gene expression was not consistently observed (one of the two

experiments). The change was small. It was reduced to 75%, while *ESR1* mRNA was reduced to 15% in the same kind of experiment (new **Fig. 5b**). Therefore, we have to avoid proposing the suggested possibility.

Reviewer #3 (Remarks to the Author):

In this manuscript, the authors investigated the long-range chromatin TADs interaction and their consequence on balance of LTED MCF-7 cells growth and apoptosis. They found lncRNA pa-Eleanor (S) mediated the interaction of two TADs. Silence of pa-Eleanor inhibited the formation of Eleanor clouds, the binding of the b site of 6q21 with ESR1 promoter and promoted LTED cell apoptosis. And inhibition of pa-Eleanor mimics the effect of resveratrol treatment. In general, this manuscript is the extension of the previous study (Nat Commun 6, 6966 (2015)) and clarified the mechanism of LTED cells primed for apoptosis. However, the roles of 6q21 TAD in promoting co-upregulation of genes in the Eleanor TAD were still unknown. Some issues need to be addressed.

Major comments:

*1. This manuscript investigated the balanced chromosomal interaction between genes for cell proliferation (*ESR1*) and death (*FOXO3*) in LTED cells, mediated by Eleanors. However, gene expression in Eleanors TAD didn't affect the activation of *FOXO3* contained 6q21 TAD. Also, there were no direct evidence to prove the interaction between the b site of 6q21 with *ESR1* promoter promoted the transcription of Eleanors and *ESR1*. It seems only pa-Eleanor regulated *ESR1* expression determined LETD cell growth or apoptosis. The authors should provide more experiments to show the functional interaction of 6q21 on *ESR1* promoter.*

We thank the reviewer for this critical comment. As pointed out, the inhibition of gene expression in the *Eleanor* TAD did not affect the expression of the *FOXO3* contained in 6q21 TAD. To investigate whether the interaction between the b site of 6q21 with the *ESR1* promoter promoted the transcription of *Eleanors* and *ESR1*, we tried to introduce chromosomal interactions between the *ESR1* promoter and 6q21 by overexpressing *pa-Eleanor(S)* in MCF7 and HeLa cells, to determine how they would affect the transcriptional levels of the genes in the Eleanor and 6q21 TAD in MCF7 cells. As shown below, we found that the overexpression did not change the expression of the genes in both chromatin domains. This may have happened, because the overexpression was exogenous and the *pa-Eleanor(S)* could not promote the chromosomal interaction. Consistently, the size of the *Eleanor* RNA cloud did not change (MCF7 cells), or the cloud was not newly formed (HeLa cells). These results are shown below.

The qRT-PCR analyses, showing that *pa-Eleanor(S)* was overexpressed in MCF7 cells (left). The expression level of the genes in the *Eleanor* TAD and 6q21 did not change upon the *pa-Eleanor(S)* overexpression in MCF7 cells (middle and right). Values of the empty plasmid-transfection were set to 1.

RNA FISH with MCF7 cells overexpressing *pa-Eleanor(S)*. The two FISH probes to detect *Eleanors* (450E24 and 63I5) were used. The genomic positions covered by the BAC clones are shown in **Fig. 1b** (green bars). The *Eleanor* RNA foci did not change with the *pa-Eleanor(S)* overexpression. Scale bar, 10 µm.

The qRT-PCR analyses, showing that *pa-Eleanor(S)* was overexpressed in HeLa cells (left). The expression level of the genes in the *Eleanor* TAD and 6q21 did not change upon the *pa-Eleanor(S)* overexpression in HeLa cells (middle and right). Values of the empty plasmid-transfection were set to 1.

RNA FISH scanning analysis along the *Eleanor* TAD, with *pa-Eleanor(S)* overexpressed in HeLa cells. Nuclear *Eleanor* RNA foci were not detected with *pa-Eleanor(S)* overexpression. Scale bar, 10 μm.

As the reviewer commented, the primary function of *pa-Eleanor(S)* could be to activate genes in *Eleanor* domains in LETD cells, which promote cell growth. As the consequence, the *Eleanor* TAD and 6q21 are co-localized and interact in the nucleus. Therefore, we have rewritten the discussion section of the manuscript on page 16, line 21.

We have also re-written the abstract and changed the title of the manuscript.

2. The group's previous paper found u-Eleanor is involved in transcriptional activation of the ESRI locus and maintains downstream intragenic Eleanor in LTED cells. The pa-Eleanor reported in the present study displayed the similar functions. The authors should discuss the relationship between u-Eleanor and pa-Eleanor. And was it a general mechanism for Eleanor ncRNAs in regulating ESRI expression?

Again, we thank the reviewer for the important comment. As this reviewer pointed out, the *pa-Eleanor(S)* reported in the present study displayed similar functions to those of *u-Eleanor*, which activates the *ESRI* locus and nearby intragenic *Eleanors* with RNA cloud formation. This may be a common mechanism for *Eleanor* ncRNAs in regulating *ESRI* mRNA expression. To understand the relationship between *u-Eleanor* and *pa-Eleanor(S)*, we performed mutual knockdown experiments. We found that the *pa-Eleanor(S)* knockdown reduced the level of *u-Eleanor*, while the *u-Eleanor* knockdown did not change the *pa-Eleanor(S)* level. This suggests that *pa-Eleanor(S)* may function upstream of *u-Eleanor*. We presented these data in the new **Supplementary Fig. 7f-h**, and described them in the revised manuscript on page 14, line 4.

Minor comments:

1. For all the cell apoptosis experiments, the authors had better provide flow cytometry results stained by annexin V/PI.

We performed the FACS analysis with Annexin V/PI staining, and the results are shown in the new manuscript (new **Supplementary Figs. 4d, g, 10a**).

2. For all the flow cytometry experiments showed in this manuscript, the authors should also provide the isotype-treatment curve for each group.

Annexin V binds directly, not through antibodies, to Phosphatidylserines (PS) exposed on the plasma membrane surface of apoptotic cells. Therefore, the conventional isotype-treatment curve is not available. Instead, we performed a FACS analysis using non-stained cells, as shown below.

These serve as appropriate negative controls for the new **Fig. 6c** and new **Supplementary Fig. 10a**.

The FACS plot image which added the non-staining cells (negative control) to **Fig. 6c** and **Supplementary Fig. 10a** were shown.

3. In figure 5a, there was interaction between c site with ESRI promoter, which was not consistent with fig 2d, 2f. the authors should explain the results.

We thank the reviewer for the careful inspection of our work. It is true that the interaction between the *ESRI* promoter and the c site was detected in the previous **Fig. 5a** (new **Fig. 6a**). However, it was weaker than the one to the b site, and was only detected in the presence of the control LNA in the previous **Fig. 5a** (new **Fig. 6a**). On the other hand, the interaction between the b site and the *ESRI* promoter was solid and observed in all of the 3C-qPCR experiments (new **Figs. 2d, 3b, and 6a**). We consider the interaction between the c site and *ESRI* to be unstable, if it is present.

4. In sup fig5e, the basal cell viability was largely different. The experimental conditions should be optimized.

According to this reviewer's comment, we re-analyzed the data. We found that the treatment of

LTED and MCF7 cells with 1 nM LNA had minimal effects on their proliferation. We then considered the cell numbers with 1 nM LNA treatment as the basal cell viability, set them at 100, and redrew the concentration-dependent growth curve in the new **Supplementary Fig. 8g**.

(References)

- Barutcu, A.R., Lajoie, B.R., McCord, R.P., Tye, C.E., Hong, D., Messier, T.L., Browne, G., van Wijnen, A.J., Lian, J.B., Stein, J.L., *et al.* (2015). Chromatin interaction analysis reveals changes in small chromosome and telomere clustering between epithelial and breast cancer cells. *Genome Biol* *16*, 214.
- Bintu, B., Mateo, L.J., Su, J.-H., Sinnott-Armstrong, N.A., Parker, M., Kinrot, S., Yamaya, K., Boettiger, A.N., and Zhuang, X. (2018). Super-resolution chromatin tracing reveals domains and cooperative interactions in single cells. *Science* *362*, eaau1783.
- Chiw, K.H., Tan, Y., Chua, R.Y., Huang, D., Ng, M.L.M., Torta, F., Wenk, M.R., and Wong, S.H. (2012). SNX3-dependent regulation of epidermal growth factor receptor (EGFR) trafficking and degradation by aspirin in epidermoid carcinoma (A-431) cells. *Cellular and Molecular Life Sciences* *69*, 1505-1521.
- Consortium, E.P. (2012). An integrated encyclopedia of DNA elements in the human genome. *Nature* *489*, 57-74.
- Donaldson-Collier, M.C., Sungalee, S., Zufferey, M., Tavernari, D., Katanayeva, N., Battistello, E., Mina, M., Douglass, K.M., Rey, T., Raynaud, F., *et al.* (2019). EZH2 oncogenic mutations drive epigenetic, transcriptional, and structural changes within chromatin domains. *Nature Genetics* *51*, 517-528.
- Ekoff, M., Kaufmann, T., Engstrom, M., Motoyama, N., Villunger, A., Jonsson, J.I., Strasser, A., and Nilsson, G. (2007). The BH3-only protein Puma plays an essential role in cytokine deprivation induced apoptosis of mast cells. *Blood* *110*, 3209-3217.
- Finn, E.H., Pegoraro, G., Brandão, H.B., Valton, A.-L., Oomen, M.E., Dekker, J., Mirny, L., and Misteli, T. (2019). Extensive Heterogeneity and Intrinsic Variation in Spatial Genome Organization. *Cell* *176*, 1502-1515.e1510.
- Gannon, F., Reid, G., Denger, S., and Koš, M. (2001). Minireview: Genomic Organization of the Human ER α Gene Promoter Region. *Molecular Endocrinology* *15*, 2057-2063.
- Haft, C.R., de la Luz Sierra, M., Barr, V.A., Haft, D.H., and Taylor, S.I. (1998). Identification of a family of sorting nexin molecules and characterization of their association with receptors. *Molecular and cellular biology* *18*, 7278-7287.
- Hao, N., Palmer, A.C., Ahlgren-Berg, A., Shearwin, K.E., and Dodd, I.B. (2016). The role of repressor kinetics in relief of transcriptional interference between convergent promoters. *Nucleic Acids Res* *44*, 6625-6638.
- Isoda, T., Moore, A.J., He, Z., Chandra, V., Aida, M., Denholtz, M., Piet van Hamburg, J., Fisch, K.M., Chang, A.N., Fahl, S.P., *et al.* (2017). Non-coding Transcription Instructs Chromatin Folding and

- Compartmentalization to Dictate Enhancer-Promoter Communication and T Cell Fate. *Cell* *171*, 103-119 e118.
- Le Dily, F., Vidal, E., Cuartero, Y., Quilez, J., Nacht, A.S., Vicent, G.P., Carbonell-Caballero, J., Sharma, P., Villanueva-Cañas, J.L., Ferrari, R., *et al.* (2019). Hormone-control regions mediate steroid receptor-dependent genome organization. *Genome Research* *29*, 29-39.
- Lieberman-Aiden, E., van Berkum, N.L., Williams, L., Imakaev, M., Ragozy, T., Telling, A., Amit, I., Lajoie, B.R., Sabo, P.J., Dorschner, M.O., *et al.* (2009). Comprehensive Mapping of Long-Range Interactions Reveals Folding Principles of the Human Genome. *Science* *326*, 289-293.
- Mazo, A., Hodgson, J.W., Petruk, S., Sedkov, Y., and Brock, H.W. (2007). Transcriptional interference: an unexpected layer of complexity in gene regulation. *J Cell Sci* *120*, 2755-2761.
- Nuebler, J., Fudenberg, G., Imakaev, M., Abdennur, N., and Mirny, L.A. (2018). Chromatin organization by an interplay of loop extrusion and compartmental segregation. *Proceedings of the National Academy of Sciences* *115*, E6697.
- Rao, S.S., Huntley, M.H., Durand, N.C., Stamenova, E.K., Bochkov, I.D., Robinson, J.T., Sanborn, A.L., Machol, I., Omer, A.D., Lander, E.S., *et al.* (2014). A 3D map of the human genome at kilobase resolution reveals principles of chromatin looping. *Cell* *159*, 1665-1680.
- Rao, S.S.P., Huang, S.C., Glenn St Hilaire, B., Engreitz, J.M., Perez, E.M., Kieffer-Kwon, K.R., Sanborn, A.L., Johnstone, S.E., Bascom, G.D., Bochkov, I.D., *et al.* (2017). Cohesin Loss Eliminates All Loop Domains. *Cell* *171*, 305-320 e324.
- Reid, G., Denger, S., Kos, M., and Gannon, F. (2002). Human estrogen receptor-alpha: regulation by synthesis, modification and degradation, Vol 59.
- Schwarzer, W., Abdennur, N., Goloborodko, A., Pekowska, A., Fudenberg, G., Loe-Mie, Y., Fonseca, N.A., Huber, W., C, H.H., Mirny, L., *et al.* (2017). Two independent modes of chromatin organization revealed by cohesin removal. *Nature* *551*, 51-56.
- Skurk, C., Maatz, H., Kim, H.S., Yang, J., Abid, M.R., Aird, W.C., and Walsh, K. (2004). The Akt-regulated forkhead transcription factor FOXO3a controls endothelial cell viability through modulation of the caspase-8 inhibitor FLIP. *J Biol Chem* *279*, 1513-1525.
- Szabo, Q., Jost, D., Chang, J.-M., Cattoni, D.I., Papadopoulos, G.L., Bonev, B., Sexton, T., Gurgo, J., Jacquier, C., Nollmann, M., *et al.* (2018). TADs are 3D structural units of higher-order chromosome organization in *Drosophila*. *Science advances* *4*, eaar8082-eaar8082.
- Tomita, S., Abdalla, M.O., Fujiwara, S., Matsumori, H., Maehara, K., Ohkawa, Y., Iwase, H., Saitoh, N., and Nakao, M. (2015). A cluster of noncoding RNAs activates the ESR1 locus during breast cancer adaptation. *Nat Commun* *6*, 6966.
- van de Werken, H.J.G., Landan, G., Holwerda, S.J.B., Hoichman, M., Klous, P., Chachik, R., Splinter, E., Valdes-Quezada, C., Öz, Y., Bouwman, B.A.M., *et al.* (2012). Robust 4C-seq data analysis to screen for regulatory DNA interactions. *Nature Methods* *9*, 969.

REVIEWERS' COMMENTS:

Reviewer #1 (Remarks to the Author):

Authors performed more experiment to answer the questions and tried to explain the concerns raised by authors.

Reviewer #2 (Remarks to the Author):

The authors have performed additional and extensive experiments to clarify several issues raised by the reviewers. They have also revised the text extensively to better explain some of their findings and conclusions. The manuscript is now appropriate for publication

Reviewer #3 (Remarks to the Author):

In this revised manuscript, the authors had well addressed all the critiques raised by the reviewers. The manuscript is now suitable for publication in NC.

Reviewer #1:

Review for NCOMM-18-19001A

>Authors performed more experiment to answer the questions and tried to explain the concerns raised by authors.

Thank you for this supportive comment. We are pleased that we accommodated the reviewer's concerns.

Reviewer #2:

>The authors have performed additional and extensive experiments to clarify several issues raised by the reviewers. They have also revised the text extensively to better explain some of their findings and conclusions. The manuscript is now appropriate for publication

We are pleased to receive this supportive comment.

Reviewer #3:

>In this revised manuscript, the authors had well addressed all the critiques raised by the reviewers. The manuscript is now suitable for publication in NC.

Thank you for this supportive comment. We are pleased that we were able to address the critiques well.